# Towards Large Reasoning Models for Agriculture

## Abstract

Agricultural decision-making involves complex, context-specific reasoning, where choices about crops, practices, and interventions depend heavily on geographic, climatic, and economic conditions. Traditional large language models (LLMs) often fall short in navigating this nuanced problem due to limited reasoning capacity. We hypothesize that recent advances in large reasoning models (LRMs) can better handle such structured, domain-specific inference. To investigate this, we introduce AGREASON, the first expert-curated open-ended science benchmark with 100 questions for agricultural reasoning. Evaluations across eighteen open-source and proprietary models reveal that LRMs outperform conventional ones, though notable challenges persist, with the strongest Gemini–based baseline achieving 36% accuracy. We also present AGTHOUGHTS, a large-scale dataset of 44.6K question-answer pairs generated with human oversight and equipped with synthetically generated reasoning traces. Using AGTHOUGHTS, we develop AGTHINKER, a suite of small reasoning models that can be run on consumer-grade GPUs, and show that our dataset can be effective in unlocking agricultural reasoning abilities in LLMs.

## 1 Introduction

Agriculture is the cornerstone of global sustenance, providing the essentials — food, feed, and fiber — needed to support growing populations (Food and Agriculture Organization of the United Nations, 2024). It is also a critical pillar of the global economy, employing nearly one-third of the world's workforce and contributing over 4% to global GDP (World Bank, 2024a;b). Decision making in agricultural settings is very nuanced; farmers depend on a variety of cues including location-specific soils, microclimates, crop varieties, and dynamic biotic threats, for everyday decision making. As a result, university extension agents and crop advisors routinely address highly contextualized questions — *"What should I do about knotweeds in my barley field located in Colorado in September?"* or *"How can I manage a low yield of corn harvest in Vermont?"* — that demand fine-grained, situational reasoning. Agricultural decision support tools based on mainstream LLMs (Yang et al., 2024b; Samuel et al., 2025) often falter in such scenarios.

Recent emergence of large reasoning models (LRMs) has shown promise in structured reasoning tasks in domains like mathematics, coding, and logic (DeepSeek-AI, 2025; Yang et al., 2025). Flagship benchmarks and reasoning-focused datasets have emerged to evaluate and fine-tune such models (Team, 2025a; White et al., 2025). However, establishing benchmarks and curating datasets for open-ended scientific decision support have been challenging, as it requires close collaboration with domain experts. Multiple choice based science benchmarks such as GPQA (Rein et al., 2023) and ScienceQA (Lu et al., 2022) are effective in evaluating broad multitask science knowledge, but do not capture the geo-spatial, seasonal, and management complexities that dominate real-world agricultural decisions. Consequently, models that perform well on these tests may still produce overly generic or impractical agronomic recommendations. Existing agriculture-specific datasets and benchmarks, such as AgXQA (Kpodo et al., 2024) and AgriBench (Zhou & Ryo, 2024) tend to focus on closed-form factual recall or narrow sub-domains. Therefore, the evaluation and fine-tuning of reasoning models in context-specific agricultural scenarios remain largely unexplored.

To address this gap, we introduce AGTHOUGHTS and AGREASON, two complementary resources aimed at advancing agricultural reasoning in language models. AGTHOUGHTS, is a dataset of 44.6K

question–answer pairs with explicit reasoning traces generated with oversight from agronomy experts. The data set contains diverse, realistic questions in ten key categories, taking into account variables such as geography, crop stage, disease risk, and weather. From this pool of questions we curate AGREASON, a benchmark of 100 open-ended questions with gold-standard responses validated by agronomy experts. These questions reflect real agricultural extension workflows and require multi-step reasoning over location-specific agronomic factors.

We evaluate 18 state-of-the-art open-source and proprietary models on AGREASON using a carefully designed LLM-as-judge to compare with expert-validated responses. Additionally, we leverage AGTHOUGHTS to produce a series of small reasoning models — AGTHINKER — to enable lightweight, domain-specific reasoning. Our results show that while LRMs significantly outperform standard LLMs on AGREASON, even the best performing LRM was only able to achieve a 36% success rate, demonstrating plenty of room for further AI progress in the agricultural domain. This also highlights that our benchmark comprises real-world agricultural reasoning questions that are truly challenging. Hence, it will also motivate model developers to deepen their understanding of agricultural reasoning. Our AGTHINKER models show significant improvement over base models and are among the best performing open-source models for agricultural domain-specific reasoning.

## 2 RELATED WORK

**Reasoning models, datasets, and benchmarks.** As LLMs grow in complexity, evaluating their reasoning capabilities using well-curated benchmarks has become essential. Recent research has focused on developing LLMs with explicit reasoning mechanisms. DeepSeek-R1 (DeepSeek-AI et al., 2025) introduced a reinforcement-learning-assisted pipeline to enhance reasoning via supervised fine-tuning (SFT). Both open-source models such as Qwen-QwQ (Team, 2025c) and LLaMA-4 (Touvron et al., 2023), and proprietary systems like OpenAI-O1, Gemini 2.5-Flash, and Claude 3.7-Sonnet demonstrate strong reasoning performance.

New datasets have emerged that capture detailed reasoning traces (see Table 1). Efforts such as SkyT1-17K (Team, 2025a) and Bespoke-Stratos-17K (Labs, 2025) provided question–response pairs enriched with intermediate reasoning steps in domains like mathematics, programming, and science. OpenThoughts-114K (Team, 2025b) expanded this effort with over 100,000 multi-domain examples used to train models like OpenThinker-32B. Similarly, Dolphin-R1 (Hartford & Cognitive Computations, 2025) contributed nearly 800,000 samples emphasizing logical inference, while GlaiveAI's Reasoning-v1-20M (AI, 2025a) introduced over 22 million examples, making it one of the largest open-domain reasoning datasets available. On the benchmarks side, Arena-Hard (Li et al., 2024) presents a curated benchmark of 500 user-sourced questions from Chatbot Arena; LiveBench (White et al., 2025) offers a dynamic, regularly updated benchmark of real-world questions across domains like math, code, reasoning, and data analysis; and GPQA Rein et al. (2024) offers challenging open-world questions focusing on basic science. A multitude of domain-specific benchmarks have also emerged; see Table B.

**Agriculture specific datasets and benchmarks.** Agriculture is a domain that demands not only factual knowledge but also context-aware reasoning. Early works, such as (Tzachor et al., 2023), highlight the limitations of GPT-style models in addressing agricultural extension questions and advocate for human-in-the-loop refinement. AGXQA (Kpodo et al., 2024) leverages fine-tuned models for domain-specific QA tasks, using human-preference evaluations to assess quality. Several datasets

Table 1: Classification of reasoning datasets for LLMs; we categorize the datasets by their size, domain of involvement, source reasoning model used in generating

| Dataset | Size | Subjects | Source Model |
|---|---|---|---|
| Sky-T1 (Team, 2025a) | 17k | Math, Code, Science | DeepSeek-R1 |
| Bespoke-Stratos-17k (Labs, 2025) | 17k | Math, Code, Puzzles | Open-source |
| medical-o1-reasoning-SFT (Chen et al., 2024) | 44.6k | Medicine | DeepSeek-R1 |
| OpenThoughts-114k (Team, 2025b) | 114k | Math, Science, Code, Puzzles | DeepSeek-R1 |
| Dolphin-R1 (Hartford & Cognitive Computations, 2025) | 800k | Math, Code, Chat | DeepSeek-R1, Gemini 2.0 |
| GlaiveAI/Reasoning-v1-20M (AI, 2025a) | 22M | Logic, Writing, Dialogue | R1-Distill-LLaMA-70B |
| **AGTHOUGHTS** (ours) | 44.6k | Science, Agriculture | DeepSeek-R1 |

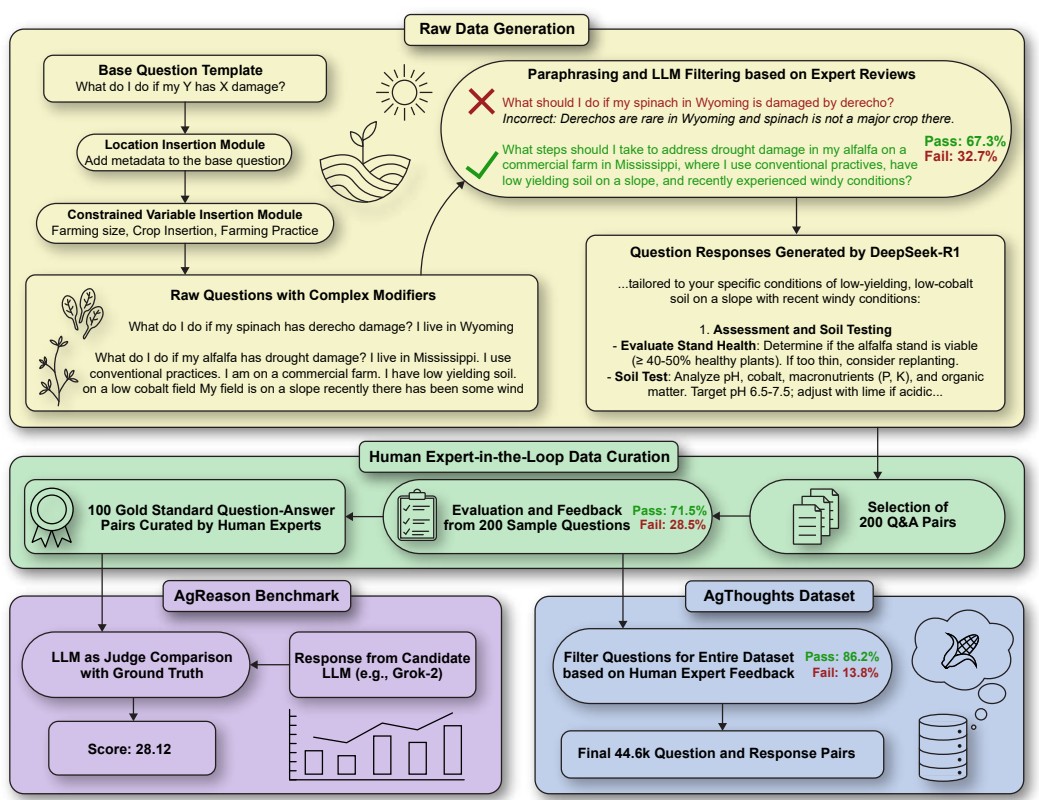

Figure 1: Workflow for the development of *AgThoughts* and *AgReason*: (1) Base templates are expanded into detailed question–answer pairs using LLMs; (2) Expert feedback on 200 sampled examples identifies common issues; (3) Expert and LLM-based feedback is used to iteratively filter and finalize 44.6k Q&A pairs; (4) The *AgReason* Benchmark evaluates candidate LLM performance using 100 questions with expert-curated gold-standard answers using LLM-as-judge.

support agricultural NLP research. The Agri-LLM raw dataset compiles processed text from agricultural documents, aiding in pretraining or fine-tuning. The KisanVaani QA dataset offers around 22.6K question-answer pairs focused on crop and soil management (AI, 2025b). AgriBench (Zhou & Ryo, 2024) and AgMMU (Gauba et al., 2025) extend this further by evaluating multimodal models through tasks combining visual and textual inputs. AgEval (Arshad et al., 2025) targets plant stress phenotyping, offering 12 diverse tasks for LLMs in zero- and few-shot settings. BioTrove (Yang et al., 2024a), a massive image dataset spanning over 366,000 species, also holds promise for integrating biological data into LLMs. Despite these contributions, there remains a gap in open-ended, reasoning-focused agricultural benchmarks designed to evaluate state-of-the-art LLMs using modern, reference-based evaluation schemes.

## 3 DATA GENERATION AND CURATION PIPELINE

Our team of agricultural domain experts carefully designed a novel question-generation workflow to ensure broad coverage across key agronomic categories. These questions were crafted to capture contextual variations, such as location, weather, and soil conditions. Initial responses (along with reasoning traces) were generated by DeepSeek-R1, then reviewed by the experts. We leverage the human feedback to create our dataset and benchmark as described below (see figure 1).

### 3.1 QUESTION GENERATION WORKFLOW

Each question in the AGTHOUGHTS dataset is constructed using a two-part schema: a base question template, and a set of modifiers that incrementally increase contextual complexity. This design enables systematic scaling of question difficulty and domain specificity.

**Base templates.** We developed a curated set of base question templates representing 10 key agronomic categories, broadly grouped into crop-related, abiotic, and biotic topics. These templates were designed to reflect a wide range of plausible field scenarios that are likely to be encountered in practical crop production contexts. The base question templates were based on common agronomic challenges and decision points—such as crop management, cover cropping, nutrient issues, in-season conditions, post-harvest concerns, soil and weather-related factors, and biotic pressures like diseases, weeds, and insect pests (see figure 2). The development process involved domain-informed generation of representative questions to ensure coverage of relevant agronomic challenges. We chose this broad range of topics to enable robust model responses across diverse situations.

**Modifiers.** To promote realism and diversity in the pool of questions, we introduced domain-specific modifiers that simulate agricultural variability. Crop modifiers were based on additional considerations used to tailor the solution to a user's specific situation. Modifier categories consisted of detail fragments concerning field conditions (soil erosion and drainage), nutrient deficiencies (lack of macro- or micronutrients in soil), planting time (economic and social considerations for planting), personal (diverse factors specific to the user affecting crop management), and weather (varying intensities of precipitation and temperature conditions) details. Modifiers were selectively included in the question formulation process, based on a probabilistic control mechanism that allows for diverse combinations.

**Question generation.** Starting from a base question template selected from one of the ten agronomic categories (see figure 2), we introduce geographic context by assigning a location relevant to agricultural variability. From there, the pipeline probabilistically selects one of two paths: (1) incorporating a crop constrained by regional conditions, or (2) layering additional modifiers (e.g., weather, soil, planting time) to further tailor the question by incorporating a crop constrained by both location and modifiers. These modifiers are sampled and combined based on domain-specific constraints, resulting in highly variable yet agronomically plausible queries. Since each question progressively increases in contextual complexity while remaining grounded in real-world agronomic scenarios by systematically varying both structural and contextual components.

**Filtering question and rephrasing.** Despite the highly constrained query generation process, the pipeline occasionally produced nonsensical or poorly formed questions. To address this, we conducted an expert review of a random sample of 200 questions from the larger set of 80,000 generated candidates. Experts evaluated the questions based on correctness, relevance, and linguistic clarity. Based on these insights, we implemented a two-step refinement process using LLMs: paraphrasing followed by filtering. We began with paraphrasing because certain question modifiers were incomplete sentences that, if added directly, would result in grammatically incorrect questions. For this step, we used the Qwen-2.5-32B-Instruct model. Following paraphrasing, we applied a custom-designed filtering prompt using the LLaMa-4-Maverick model to remove questions that still failed to meet quality standards. After this filtering stage, we retained approximately 53,860 questions, which comprised around 67.3% of the original candidate set.

### 3.2 RESPONSE GENERATION AND CURATION

We used DeepSeek-R1 to generate responses and reasoning traces for 51,800 questions ($\approx$ 2000 questions were skipped due to API response issues). To evaluate the validity of these synthetic responses, we adopted a hybrid validation scheme involving human experts, followed by an LLM judge to filter wrong or irrelevant responses from our corpus. We describe our methodology below.

**Human evaluation.** To better understand the trends of failures in the synthetic response generation by the reasoning model, we sampled 200 question–answer (Q-A) pairs from our corpus of 51,800 samples. We had 10 domain experts (advanced graduate students in the Agronomy department at Iowa State) manually reviewed 200 Q-A pairs (approximately 20 Q-A pairs each). Each pair was evaluated across three main dimensions: the question, the model's reasoning steps, and the final answer. Questions were assessed for clarity, relevance, and whether they reflected plausible agronomic

scenarios. Reasoning was evaluated for logical coherence and factual accuracy of intermediate statements. Answers were reviewed for correctness, relevance to the core problem, and inclusion of context-appropriate agronomic recommendations. In addition to marking each dimension as correct or incorrect, experts provided open-ended comments explaining their assessments. These annotations included rationale for errors, suggestions for improving response quality, and links to scientific literature or agricultural extension resources. Reviewers were explicitly instructed not to use AI assistance during this process. Evaluation was conducted using the Label Studio (Tkachenko et al., 2020-2025) platform (see Appendix). Experts dedicated between 30 to 60 minutes per Q-A pair, depending on complexity. To ensure consistency and reduce subjectivity, regular calibration meetings were held throughout the process, where reviewers discussed ambiguous cases and refined shared evaluation guidelines. Out of 200 responses, 57 were labeled incorrect. The expert comments were summarized to develop an error taxonomy for the incorrect responses (see Table C). Primary issues included factual inaccuracies, incomplete agronomic recommendations, and mismatches between the response and the contextual details of the question.

Following the primary review, evaluators revisited a subset of 100 high-quality responses to extract essential content elements: key agronomic facts or reasoning steps necessary for an answer to be considered high quality. These formed the AGREASON benchmark dataset (section 4).

**LLM-based response filtering.** To enhance the reliability of the final Q-A dataset (AGTHOUGHTS), we employed an automated filtering phase using LLM to systematically evaluate and remove flawed responses. The LLM filter was designed using the error taxonomy described above. We first used GPT-4o to analyze the open-ended expert annotations and synthesize a structured set of failure modes that could inform automated evaluation. These patterns were formalized into a rubric that defined five key dimensions of response quality: factual accuracy, contextual relevance, practical feasibility, logical consistency, and completeness. This rubric was then embedded into a custom evaluation prompt (see Appendix) for GPT-4.1, which we used as an LLM-based filter.

Each Q-A pair was independently assessed by the GPT-4.1 based filter, where the model was instructed to reason explicitly about the presence of common errors and score responses across the various criteria. These individual scores were aggregated into a composite correctness score, which was then compared against a pre-defined threshold to determine whether the response should be retained or filtered out.

## 4 DATASET, BENCHMARK, AND REASONING MODELS

**The AGTHOUGHTS Dataset.** Based on the filtering process described above, we assemble the AGTHOUGHTS dataset. This contains 44,600 curated questions, each paired with detailed answers and reasoning traces from DeepSeek-R1, totaling around 66.2 million tokens. On average, questions are 26 words long, answers span 354 words, and reasoning traces are 725 words long. The dataset covers a wide range of agronomic topics. The dataset spans major agronomical categories and specialized areas, as reflected in the distribution of questions - Plant and seed health questions are the most frequent (15,811), followed by crop management (7,539), general management (6,105), harvest (4,139), soil (3,896), and weather (3,527). Cover cropping appears in 1,969 questions. Biotic categories include diseases (819), insects (702), and weeds (108). This breadth and depth make AGTHOUGHTS a rich resource for evaluating complex agricultural reasoning. See figure 2 for example questions for each of these categories.

**The AGREASON Benchmark.** As mentioned above, after primary review we assemble the AGREASON benchmark. This comprises 100 Q-A pairs (10 from each category) with high-quality answers that were set aside during human evaluation as mentioned earlier. The wording of the responses (originally produced by DeepSeek-R1) were further refined by the experts to generate gold-standard answers for the benchmark.

**LLM-as-Judge design for benchmark evaluation.** To rigorously assess both the *correctness* and *completeness* of the answers generated by an LLM under evaluation, we adopt a fine-grained, statement-level evaluation protocol mirroring the workflow proposed in FACTS Grounding (Jacovi et al., 2025). For each test question in our benchmark, the "LLM-as-a-Judge" pipeline is provided with the original user query, the corresponding gold standard answer from the benchmark, and a

candidate response generated by the model under evaluation (see Appendix for the full evaluation prompt). The judge model then processes the candidate response by decomposing it into individual statements. Each statement which is aligned with statements in the gold standard answer is labeled as either *supported*, *unsupported*, or *contradictory*. While the *supported* category indicates a semantic match, *unsupported* and *contradictory* refer to mismatched and conflicting facts respectively. Additionally, any relevant fact present in the gold standard answer but missing from the candidate response is marked as a *missing fact*. Based on this evaluation, we define the following statement categories.

- **True Positives (TP):** statements correctly stated and labeled *supported*.
- **False Positives-unsupported ($FP_u$):** statements included but labeled *unsupported*.
- **False Positives-contradictory ($FP_c$):** statements included but labeled *contradictory*.
- **False Negatives (FN):** Facts in gold standard answers missing from the candidate response.

Then, we compute standard metrics—precision, recall, and F1-score for each question based on the above definitions of TP, FP, and FN (see Appendix for details). To determine when a model response is deemed *acceptable*, we conducted expert reviews on a subset of the benchmark questions and identified an F1-score threshold that aligns closely with human judgment of the answer quality. A response is labeled as a *pass* if its F1-score is above this threshold. We then calculate the overall pass rate for each model, defined as the percentage of benchmark questions for which the model achieves a passing F1-score. This pass rate serves as the primary metric for model comparison based on the benchmark.

**AGTHINKER models.** We perform supervised fine-tuning (SFT) on the entire AGTHOUGHTS dataset to unlock agronomy-specific reasoning in smaller language models. our approach involves

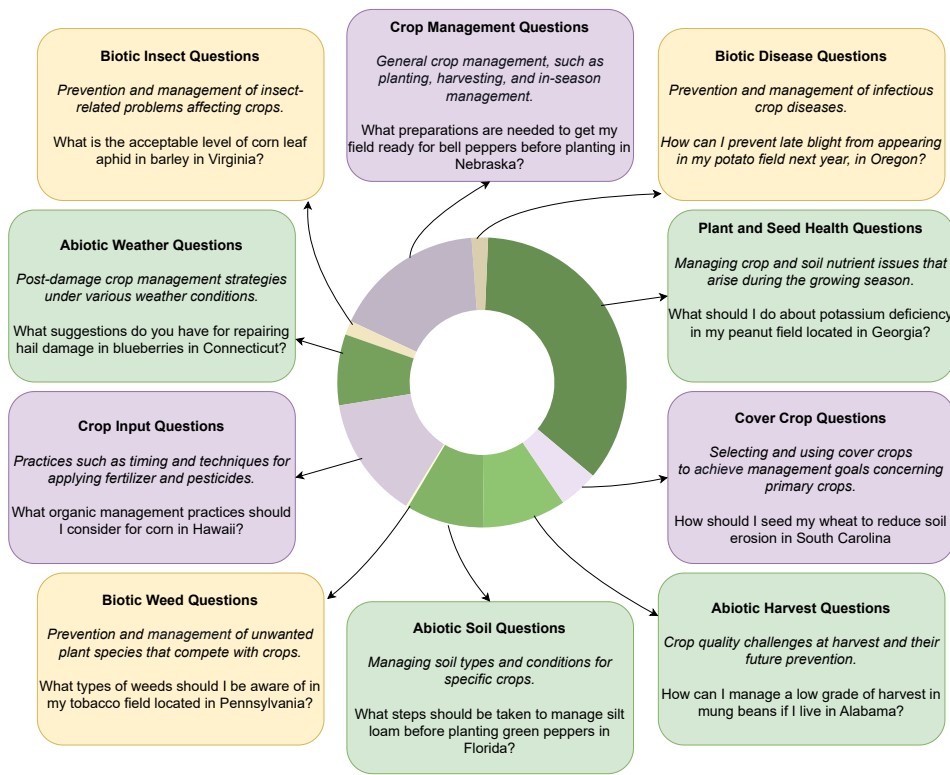

Figure 2: Ten expert-defined agronomic question categories used in AGTHOUGHTS and AGREA-SON.

three stages, and is inspired by the pipeline proposed in *OpenThoughts* Team (2025b), which demonstrated the efficacy of trace-augmented examples for reasoning transfer. We first filter the dataset to exclude any overlap with benchmark questions to prevent evaluation leakage. Then we format each instance, which includes a multi-step reasoning trace, and a final answer, into a dialogue-style input augmented with special tokens (<think> and </think>) to mark reasoning segments. We fine-tune a range of open-source base models using full supervised fine-tuning, where all model parameters are updated during training. Specifically, our suite includes small models (Phi-3 3B, Qwen-2.5 3B), medium models (Mistral-7B, Qwen-2.5 8B, LLaMA-3 8B, Qwen-2.5 14B, Phi-3 14B). This suite of fine-tuned models, that we collectively call AGTHINKER, is designed to distill reasoning capabilities from larger models into smaller, domain-adapted architectures. Evaluation on our AGREASON benchmarks shows that AGTHINKER models consistently outperform their base counterparts, validating the effectiveness of our fine-tuning pipeline for agricultural decision support.

## 5 EXPERIMENTAL RESULTS

In this section, we present the outcomes of our evaluation of a range of open-source and proprietary LLMs on our benchmark. Models are assessed based on the percentage of responses that achieve an F1 score above the threshold of 0.8. This threshold was determined via the following expert review process. We presented experts with model responses at varying F1 score levels and identified the level at which the answers were consistently judged to be complete and satisfactory. figure 3 illustrates how the percentage of answers rated above a given threshold changes as the threshold varies.

### 5.1 BENCHMARK RESULTS

Our evaluation shows that generally, reasoning-focused models consistently outperform conventional LLMs (Table 8). This supports the notion that answering questions in our benchmark requires reasoning steps to tailor the responses to the specific context of each query. To ensure representative evaluation, we include at least one model from each major vendor. Among them, **Grok-3 beta** and **Gemini 2.5 Flash** achieved the highest recall scores of **0.815** and 0.778, respectively, demonstrating strong effectiveness in capturing true positives. Notably, **Gemini 2.5 Flash** also achieved the best precision (**0.727**) and the highest overall score (**36%**). In contrast, models such as **Mistral-24B** and **LLaMA-4 Scout** exhibited lower performance, with both precision and recall below 0.55. **GPT OSS 20B**, a recent open-source model from OpenAI, reached **9%**.

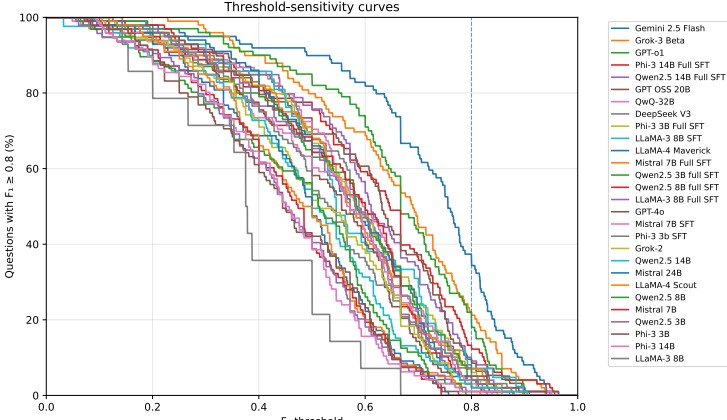

Figure 3: Percentage of questions with F1 scores above varying thresholds.

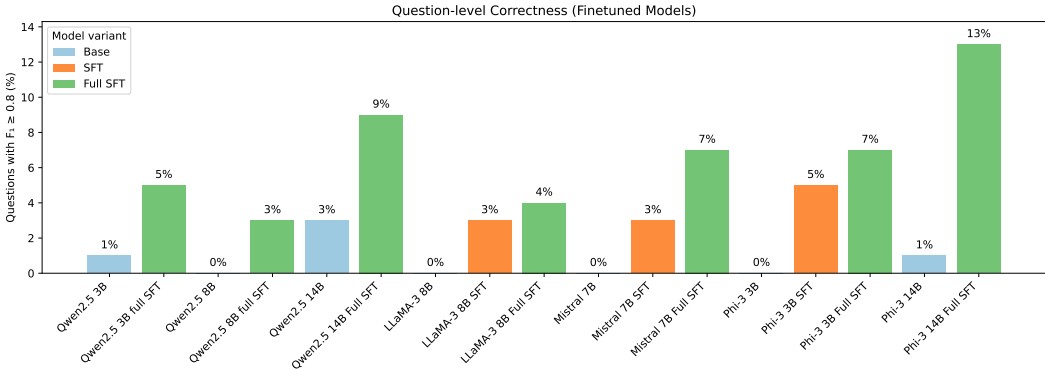

Figure 4: Question-level pass rate ($F_1 \geq 0.8$) comparing base (before fine-tuning) models with AG-THINKER models (post-SFT). **Full SFT** indicates full fine-tuning of all model parameters, whereas **SFT** refers to parameter-efficient fine-tuning using the LoRA approach.

| Model | Score (F$_1$>0.80) | Precision | Recall |
|---|---|---|---|
| Gemini 2.5 Flash | 36.0% | 0.727 | 0.778 |
| Grok-3 Beta | 22.0% | 0.583 | 0.815 |
| GPT-o1 | 20.0% | 0.654 | 0.710 |
| Phi-3 14B Full SFT | 13.0% | 0.564 | 0.719 |
| Qwen2.5 14B Full SFT | 9.0% | 0.560 | 0.681 |
| GPT OSS 20B | 9.0% | 0.534 | 0.731 |
| Mistral 7B Full SFT | 7.0% | 0.526 | 0.628 |
| DeepSeek V3 | 7.0% | 0.544 | 0.644 |
| Phi-3 3B Full SFT | 7.0% | 0.524 | 0.661 |
| Phi-3 3b SFT | 5.0% | 0.474 | 0.598 |
| Qwen2.5 3B Full SFT | 5.0% | 0.514 | 0.658 |
| QwQ-32B | 5.0% | 0.505 | 0.693 |
| GPT-4o | 5.0% | 0.554 | 0.558 |
| LLaMA-3 8B Full SFT | 4.0% | 0.518 | 0.622 |
| LLaMA-4 Maverick | 4.0% | 0.596 | 0.593 |
| Mistral 7B SFT | 3.0% | 0.470 | 0.678 |
| Qwen2.5 8B Full SFT | 3.0% | 0.503 | 0.644 |
| Qwen2.5 14B | 3.0% | 0.515 | 0.533 |
| LLaMA-3 8B SFT | 3.0% | 0.372 | 0.399 |
| Grok-2 | 3.0% | 0.466 | 0.575 |
| Qwen2.5 3B | 1.0% | 0.422 | 0.501 |
| LLaMA-4 Scout | 1.0% | 0.480 | 0.523 |
| Phi-3 14B | 1.0% | 0.548 | 0.400 |
| Qwen2.5 8B | 0.0% | 0.457 | 0.513 |
| Mistral 24B | 0.0% | 0.442 | 0.557 |
| LLaMA-3 8B | 0.0% | 0.183 | 0.088 |
| Mistral 7B | 0.0% | 0.408 | 0.520 |
| Phi-3 3B | 0.0% | 0.440 | 0.452 |

Table 2: Per-model benchmark performance. Rows shaded in peach denote reasoning models, while those shaded in cyan and gray indicate AGTHINKER models. **Full SFT** indicates full fine-tuning of all model parameters, whereas **SFT** refers to parameter-efficient fine-tuning using the LoRA approach.

## 5.2 CATEGORY-BASED RESULTS

To further analyze model performance across diverse agronomic tasks, we evaluated the pass-rate scores for each model across ten distinct question categories, as illustrated in figure 5. The radar plot reveals that models such as **Gemini 2.5 Flash** and **Grok-3 Beta** outperform others across most categories, particularly excelling in areas such as *Biotic Diseases* and *Abiotic Soil*. In contrast, models like **DeepSeek V3**, **GPT-4o**, and **Mistral-24B** show limited effectiveness, with near-zero scores

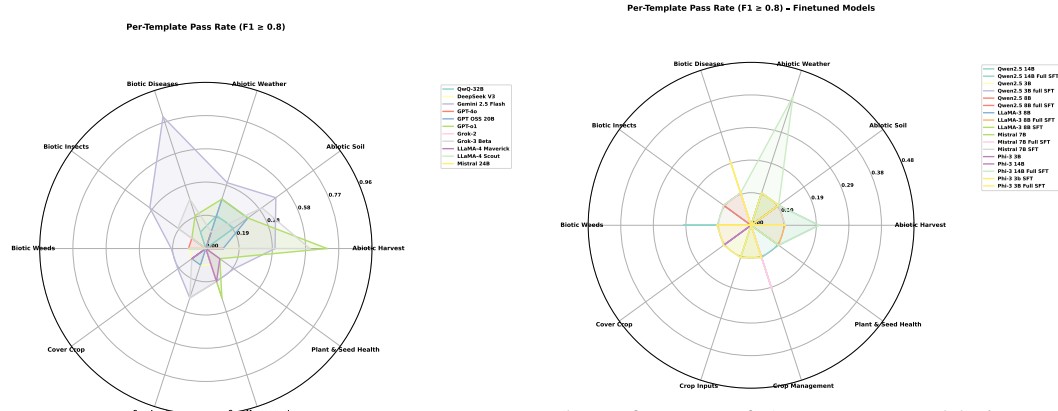

(a) Performance of models across each category.

(b) Performance of AGTHINKER models in categories.

Figure 5: Performance of models across each question category. Category-level pass rates ($F_1 \geq 0.8$) highlight the superior performance of reasoning models.

in several categories. Our analysis confirms the overall superiority of certain advanced models and points out specific strengths. Notable points include **o1**'s strong performance on *Abiotic Harvest*, and the general difficulty of *Cover Crop* where most models consistently under-perform. To summarize, these results show that current state-of-the-art LLMs have considerable room to improve in terms of accurately solving complex agronomic reasoning tasks.

### 5.3 IMPACT OF SUPERVISED FINE-TUNING ON MODEL PERFORMANCE

To evaluate the effectiveness of our fine-tuning approach, we compared the base models with their AGTHINKER counterpart. As shown in figure 5b and figure 4, fine-tuning resulted in consistent improvements across nearly all question categories. Notably, categories such as *Abiotic Weather* and *Abiotic Harvest* showed substantial gains, improving from zero or minimal scores in the base models to significant values after fine-tuning. Table 8 further supports these findings: **Phi-3 14B** achieved a score improvement from 1% to 13%, outperforming open source reasoning models like **GPT OSS 20B**.

## 6 DISCUSSION

In this work, we introduced AGTHOUGHTS and AGREASON, two complementary resources designed to advance agricultural reasoning in large language models. In collaboration with agronomy experts we developed a structured pipeline to generate realistic, context-rich questions and answers, addressing the multifactorial nature of on-farm decision-making. AGTHOUGHTS provides a first-of-its-kind dataset of over 44K expert-guided agricultural Q-A pairs with reasoning traces, while AGREASON serves as a gold-standard benchmark of 100 scenario-based fine-grained agronomy questions Compared with other reasoning benchmarks (e.g., **SIME-24/25** with 30 questions or **GPQA** with a few hundred), our **100-question** benchmark fits well within this typical scale for robust evaluation. We evaluated several state-of-the-art models and demonstrated that large reasoning models (LRMs) consistently outperform standard LLMs. The low performance even from the state-of-the-art reasoning models demonstrates the need for potentially domain-specific models. We fine-tuned compact domain-specific models (that we call AGTHINKER) which provide competitive performance with low computational overhead. With a **12%** improvement from Full SFT fine-tuning AGTHINKER - **Phi-3 14B** demonstrates the best performance among all the currently open sourced models, outperforming **GPT OSS 20B** and **DeepSeek-V3**. This shows the effectiveness of both the dataset and the fine-tuning process. In future work, we aim to expand the size of the benchmark and incorporate additional modalities to support more comprehensive, multimodal agricultural reasoning.

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

# A APPENDIX

# A COMPONENTS USED IN AGTHOUGHTS QUESTION GENERATION

## A.1 GEOGRAPHIC CONTEXTS

To reflect real-world diversity in agricultural conditions, our dataset incorporates a wide range of geographic contexts. Currently, these include all the U.S. states with varying climates, soil types,

and crop suitability. In future, we aim to broaden the scope to include locations outside the United States.

**Locations**: *Alabama, Alaska, Arizona, Arkansas, California, Colorado, Connecticut, Delaware, Florida, Georgia, Hawaii, Idaho, Illinois, Indiana, Iowa, Kansas, Kentucky, Louisiana, Maine, Maryland, Massachusetts, Michigan, Minnesota, Mississippi, Missouri, Montana, Nebraska, Nevada, New Hampshire, New Jersey, New Mexico, New York, North Carolina, North Dakota, Ohio, Oklahoma, Oregon, Pennsylvania, Rhode Island, South Carolina, South Dakota, Tennessee, Texas, Utah, Vermont, Virginia, Washington, West Virginia, Wisconsin, Wyoming*

## A.2 CROP SELECTION

During question generation, crops were selected based on a probabilistic decision to include additional complexity or not. Additional complexity was introduced by mentioning factors such as Farming Practice (conventional or organic), Farm Size (small or large), and other modifiers. When such complexity was included, crop selection was constrained by these factors along with location. In the general case, when no additional complexity was mentioned, crop selection was based solely on location. This constrained selection was guided by expert input. The number of crops used in question generation is shown in Table 3.

Table 3: Number of crops used in query generation for all constrained cases. The "General" column represents cases with minimal additional complexity. The other columns correspond to cases involving (Farming Practice, Farm Size).

| State | General | Conventional, Large Farm | Conventional, Small Farm | Organic, Large Farm | Organic, Small Farm |
|---|---|---|---|---|---|
| Alabama | 51 | 9 | 37 | 1 | 0 |
| Alaska | 16 | 5 | 9 | 0 | 1 |
| Arizona | 33 | 17 | 17 | 13 | 5 |
| Arkansas | 16 | 6 | 9 | 1 | 1 |
| California | 56 | 40 | 7 | 19 | 1 |
| Colorado | 35 | 16 | 15 | 9 | 3 |
| Connecticut | 25 | 17 | 6 | 2 | 2 |
| Delaware | 22 | 16 | 4 | 3 | 0 |
| Florida | 43 | 18 | 8 | 3 | 1 |
| Georgia | 49 | 19 | 23 | 6 | 4 |
| Hawaii | 8 | 5 | 2 | 1 | 1 |
| Idaho | 52 | 19 | 21 | 11 | 8 |
| Illinois | 50 | 12 | 31 | 5 | 7 |
| Indiana | 50 | 11 | 31 | 4 | 7 |
| Iowa | 49 | 9 | 31 | 3 | 11 |
| Kansas | 54 | 9 | 33 | 4 | 9 |
| Kentucky | 33 | 7 | 22 | 2 | 1 |
| Louisiana | 16 | 8 | 7 | 1 | 2 |
| Maine | 25 | 9 | 14 | 6 | 1 |
| Maryland | 25 | 6 | 17 | 4 | 4 |
| Massachusetts | 25 | 7 | 16 | 4 | 5 |
| Michigan | 59 | 15 | 28 | 10 | 7 |
| Minnesota | 59 | 15 | 29 | 10 | 7 |
| Mississippi | 41 | 11 | 25 | 0 | 7 |
| Missouri | 52 | 8 | 37 | 4 | 10 |
| Montana | 43 | 15 | 18 | 8 | 2 |
| Nebraska | 59 | 15 | 28 | 9 | 8 |
| Nevada | 33 | 10 | 20 | 3 | 6 |
| New Hampshire | 25 | 7 | 16 | 4 | 4 |
| New Jersey | 25 | 7 | 16 | 2 | 4 |
| New Mexico | 33 | 14 | 16 | 8 | 4 |
| New York | 43 | 9 | 23 | 6 | 9 |
| North Carolina | 38 | 9 | 23 | 5 | 11 |
| North Dakota | 17 | 8 | 6 | 5 | 5 |
| Ohio | 59 | 6 | 43 | 3 | 13 |
| Oklahoma | 32 | 6 | 23 | 2 | 6 |
| Oregon | 36 | 11 | 22 | 8 | 9 |
| Pennsylvania | 44 | 20 | 14 | 9 | 1 |
| Rhode Island | 25 | 18 | 6 | 2 | 0 |
| South Carolina | 52 | 17 | 27 | 1 | 5 |
| South Dakota | 17 | 7 | 9 | 3 | 2 |
| Tennessee | 33 | 5 | 17 | 2 | 5 |
| Texas | 32 | 12 | 16 | 5 | 9 |
| Utah | 33 | 11 | 21 | 8 | 2 |
| Vermont | 25 | 13 | 8 | 1 | 0 |
| Virginia | 33 | 11 | 19 | 3 | 10 |
| Washington | 36 | 16 | 16 | 9 | 7 |
| West Virginia | 33 | 9 | 20 | 1 | 5 |
| Wisconsin | 59 | 19 | 28 | 6 | 9 |
| Wyoming | 35 | 14 | 12 | 10 | 0 |

### A.3 AGRONOMIC MODIFIERS

To introduce realistic complexity, we use expert-guided, farm-size-informed modifiers that reflect the types of details farmers and advisors consider. These include:

**Field Conditions**: "*My field has moderately deep soil*", "*My field has poorly drained soil*", "*My field has uneroded soil*"

**Weather Details**: "*recently there was a derecho*", "*this year has been cold*", "*we have been having hail for a long time*"

**Nutrient Deficiencies**: "*on a low magnesium field*", "*on a low phosphorous field*", "*my field has high molybdenum*"

**Personal**: "*I don't have much experience with this.*", "*My neighbor has the same problem.*", "*I will be traveling next week.*"

**Planting Time**: "*I planted later this year than I normally do*", "*I planted before my insurance date*", "*My neighbor wanted me to plant his field, so I planted my field early this year*"

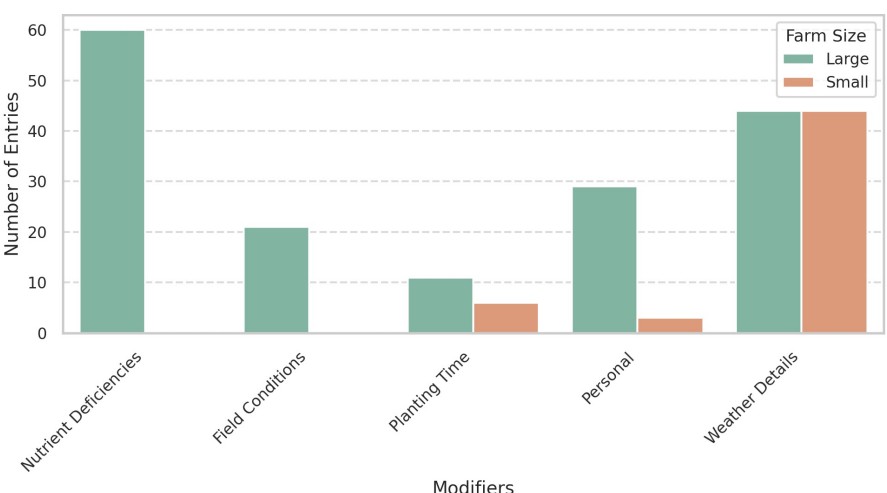

Figure 6: Distribution of modifiers by farm size.

The distribution of modifiers based on farm size is illustrated in figure 6

## B REASONING BENCHMARKS

Table 4: Classification of reasoning benchmarks for LLMs; we categorize benchmarks on domain of dataset, whether human experts were involved in the data curation process, whether the evaluation tasks are ground-truth or open-ended, and the types of the task involved.

| Benchmark | Domain | Human-in-the-loop | Open-ended | Question Type |
|---|---|---|---|---|
| LiveCodeBench (Jain et al., 2024) | Coding | Yes | No | Code Generation/Completion |
| LiveBench (White et al., 2025) | Math, Coding | Yes | No | Multiple Choice/Short Answer |
| MedXpertQA (Zuo et al., 2025) | Medicine | No | No | Multiple Choice |
| LegalBench (Guha et al., 2023) | Law | Yes | Partial | Multiple Choice/Free Response |
| ScienceQA (Lu et al., 2022) | Science | No | No | Multiple Choice |
| GPQA (Rein et al., 2023) | Science | Yes | No | Multiple Choice |
| **AGREASON** (ours) | Agriculture | Yes | Yes | Free Response |

# C  DISTRIBUTION OF ERRORS

Table 5: Error taxonomy based on human evaluation

| A. Distribution of Incorrect Answer Types | |
|---|---|
| Answer is factually wrong | 55.1% |
| Answer is cut-off | 28.6% |
| Missing common suggestion | 14.8% |
| Does not address core question | 4.1% |
| **B. Distribution of Common Error Patterns** | |
| Generalizing fertilizer or pesticide use without soil test | 13.9% |
| Inaccurate crop-specific recommendations | 11.9% |
| Assumptions without clarifying info | 11.6% |
| Ignoring economic or labor feasibility | 10.9% |
| Misunderstanding crop lifecycle/regional fit | 10.3% |
| Irrelevant/confusing to question intent | 9.9% |
| Incorrect pest/weed behavior attribution | 9.3% |
| Improper agronomic timing/sequencing | 8.9% |
| Outdated or banned chemical advice | 7.3% |
| Conflicting internal logic | 6.0% |

# D  ANNOTATION INTERFACE EXAMPLE

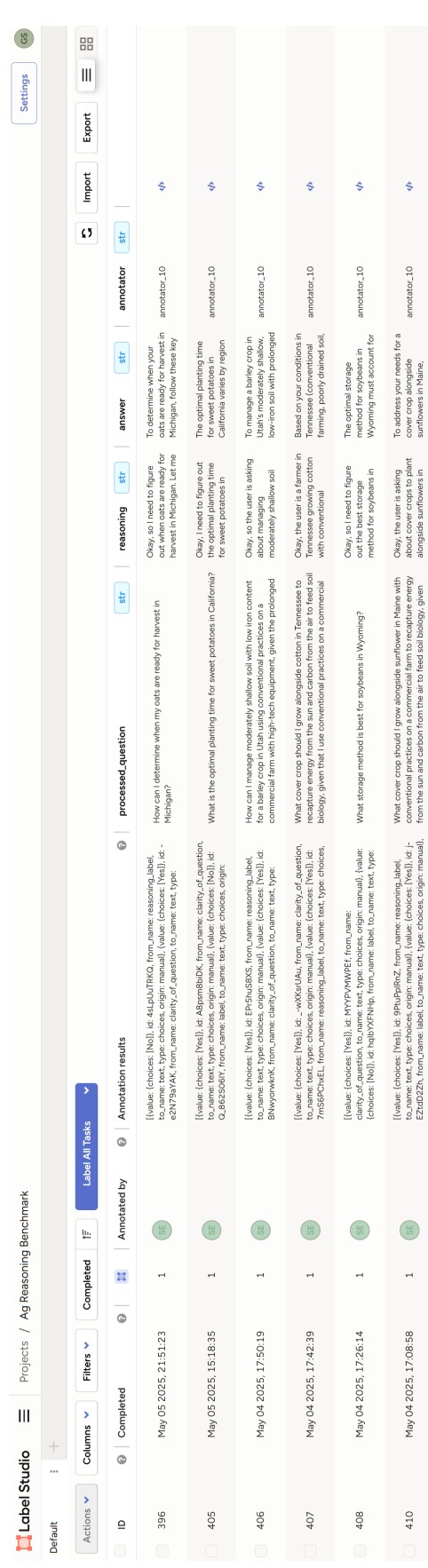

Figure 7: Screenshot of Label Studio interface illustrating the annotation workflow used by experts.

**UI Preview**

📝 **Question**

Sample: Your text will go here.

🧠 **Reasoning**

Sample: Your text will go here.

🔍 Is the reasoning for this question accurate?

☐ Yes[1]

☐ No[2]

✅ **Answer**

Sample: Your text will go here.

🔍 Is the objective of the question clear and is the content applicable?

☐ Yes[3]

☐ No[4]

💬 **Brief comment on the question**

Enter a brief comment regarding the question

Add

✔️ Is the answer to the question correct?

☐ Yes[5]

☐ No[6]

🔍 Is the answer easy to understand?

☐ Yes[7]

☐ No[8]

💬 **Brief comment on the answer**

Enter a brief comment regarding the answer

Add

Figure 8: Screenshot of the Label Studio interface illustrating the expert annotation workflow. The interface displays the question, reasoning trace, and final answer under clearly labeled sections. Experts were asked to evaluate the correctness of each component and provide comments or relevant links in case of errors.

# E   COMMON ERROR PATTERNS IN AGTHOUGHTS RESPONSES

Below are the common LLM (DeepSeek-R1) error patterns found in responses within the AGTHOUGHTS dataset. These examples are obtained from expert review and filtered using GPT-4o. Each box shows the error pattern, the expert comment that flagged it, and the original question that exposed the flaw.

---

**Error Pattern:** *Generalizing fertilizer/pesticide use without soil test*

**Example Question:** *What suggestions do you have for improving the low yield of kale on a small farm in Maryland using conventional practices, given that flooding is occurring and I will be traveling next week?*

**Response:**
To address low kale yield exacerbated by flooding on your Maryland farm, here's a structured plan combining immediate actions and arrangements for your absence:

**Immediate Actions (Before Traveling)**

1. **Drainage Improvement:**

   - **Create Ditches**: Dig shallow trenches around kale beds to divert excess water.
   - **Add Organic Matter**: Mix compost or aged manure into soil to improve structure (if soil isn't fully waterlogged).
   - **Sand Application**: Sprinkle coarse sand around plants to enhance drainage temporarily.

2. **Disease  Pest Control:**

   - **Fungicides**:  Apply copper-based fungicides to combat root rot/downy mildew.
   - **Pesticides**: Use slug bait (iron phosphate) and insecticidal soap for aphids/slugs.
   - **Remove Affected Plants**: Prune damaged leaves and uproot severely infected plants.

3. **Soil Nutrition:**

   - **Balanced Fertilizer**: Apply a granular NPK (10-10-10) with calcium to replenish nutrients lost to flooding.
   - **Foliar Spray**: Use a liquid fertilizer (e.g., seaweed extract) for quick nutrient uptake.

4. **Weed  Moisture Management:**

   - **Mulching**: After water recedes, apply straw mulch to suppress weeds and retain soil moisture.

**During Your Absence**

1. **Delegate Monitoring:**

   - Arrange for a trusted person to check fields weekly for waterlogging, pests, and disease. Provide clear instructions for emergencies (e.g., reapplying fungicides).

2. **Slow-Release Fertilizer:**

   - Apply a slow-release granular fertilizer to sustain nutrient levels.

3. **Automate Drainage:**

   - Install temporary French drains or pump systems if flooding persists.

**Long-Term Strategies (Post-Travel)**

1. **Raised Beds**: Construct permanent raised beds to prevent future flooding.

2. **Soil Testing**: Conduct a pH/nutrient test to guide amendments (target pH 6.0–7.5).

---

3. **Resistant Varieties**: Plant flood-tolerant kale varieties (e.g., 'Winterbor' or 'Red Russian').

4. **Cover Crops**: Sow clover or rye post-harvest to improve soil health.

5. **Crop Rotation**: Rotate with non-brassicas (e.g., legumes) to break pest/disease cycles.

**Additional Tips**

- **Weather Monitoring**: Use apps like FarmWise to track rainfall and adjust plans.

- **Shade Cloth**: Deploy if heat stress follows flooding to protect plants.

By implementing these steps, you can mitigate immediate flooding impacts and establish resilience for future seasons. Safe travels!

**Expert Comment:** The specific fertilizer that is used is not necessarily 10-10-10 and will vary based on soil test results. Follow up with a soil test result, although soil is not the primary issue and is not really mentioned in the question at all.

---

**Error Pattern: *Inaccurate crop-specific recommendations***

**Example Question:** *How can I save my spinach crop from drought damage in Tennessee?*

**Response**:
To protect your spinach crop from drought damage in Tennessee, implement the following strategies, organized by effectiveness and practicality:

**1. Optimize Irrigation**

- **Drip Irrigation or Soaker Hoses**: Install these systems to deliver water directly to the roots, minimizing evaporation. Spinach's shallow roots benefit from consistent moisture. Water early morning or late evening.

- **Watering Schedule**: Aim for 1-1.5 inches of water weekly. Check soil moisture daily during drought; water when the top inch is dry. Adjust to shallow, frequent watering if soil dries quickly.

**2. Mulch Application**

- **Organic Mulch**: Apply 2-3 inches of straw, compost, or untreated grass clippings around plants. Avoid stem contact to prevent rot. Mulch after watering to retain moisture and cool soil.

**3. Soil Improvement**

- **Enhance Organic Matter**: Incorporate compost or well-rotted manure to improve water retention. Raised beds can help manage moisture but monitor for drying.

**4. Shade Management**

- **Shade Cloth (30-50%)**: Use a temporary structure to shield plants from midday sun. Ensure 4-6 hours of sunlight daily to maintain growth without stress.

**5. Drought-Resistant Varieties**

- **Select Cultivars**: Choose varieties like 'Bloomsdale Long Standing' or 'Tyee' for better drought tolerance and slower bolting.

**6. Adjust Planting Schedule**

- **Cool-Season Planting**: Grow spinach in early spring or late summer/fall to avoid peak heat. Use row covers to extend seasons and reduce evaporation.

**7. Monitor and Adapt**

- **Check Soil Moisture**: Use a meter or finger test. Watch for wilting or bolting, which signal stress.

- **Rainwater Harvesting**: Collect rainwater in barrels for irrigation during dry spells.

**8. Reduce Stressors**

- **Avoid Fertilizers**: Especially high-nitrogen types, which can stress plants. Use slow-release options if necessary.
- **Windbreaks**: Plant taller crops or install barriers to reduce wind-driven evaporation.

**9. Emergency Measures**

- **Container Gardening**: In severe drought, grow spinach in self-watering containers placed in shaded areas.

**10. Bolting Prevention**

- **Consistent Moisture and Shade**: These reduce heat stress, delaying bolting. Harvest leaves promptly to encourage growth.

**Regional Consideration:**

- Tailor practices to your area in Tennessee; mountainous regions may retain moisture longer, while western areas might need more aggressive irrigation.

By integrating these methods, you can mitigate drought impact, ensuring a healthier spinach crop with sustained yield.

**Expert Comment:** Choose varieties like 'Bloomsdale Long Standing' or 'Tyee' for better drought tolerance and slower bolting. I would not give a specific variety, since breeding will release new varieties continuously.

---

**Error Pattern:** *Assumptions without asking for clarifying information*

**Example Question:** *What cover crop should I grow with strawberries in Oregon to manage excess soil moisture, using organic practices on a small farm, considering early season fog and that I planted earlier than my neighbors?*

**Response:**
To manage excess soil moisture in your Oregon strawberry farm using organic practices, consider the following cover crop strategies tailored to your conditions (early season fog, early planting, small-scale):

**1. White Clover (Living Mulch)**

- **Benefits**: Low-growing, fixes nitrogen, improves soil structure, suppresses weeds, and tolerates shade. Enhances biodiversity without competing heavily with strawberries.
- **Considerations**: Ensure it's mowed regularly to prevent excessive moisture retention. Avoid dense stands to balance moisture management and airflow.
- **Timing**: Sow in early spring or late summer between strawberry rows.

**2. Annual Ryegrass (Between Rows)**

- **Benefits**: Deep roots absorb excess water, improve drainage, and prevent erosion. Fast-growing and easy to terminate.
- **Considerations**: Mow periodically to avoid competition. Plant in fall after harvest or early spring between rows.
- **Timing**: Ideal for wet seasons; terminate before strawberries peak.

**3. Daikon Radish (Winter Cover Crop)**

- **Benefits**: Taproots break up compacted soil, enhance infiltration, and winter-kill, leaving organic channels for drainage.
- **Timing**: Plant in late summer/fall post-harvest. Decomposes by spring, prepping soil for strawberries.

**Expert Comment:** The model should ask a clarifying question about how the cover crop is planted. Is the cover crop planted between strawberry rows or after termination of the strawberry crop?

**Error Pattern:** *Ignoring economic or labor feasibility*

**Example Question:** *What suggestions do you have for improving the low yield of kale on a small farm in Maryland using conventional practices, given that flooding is occurring and you will be traveling next week?*

**Response**:

To address low kale yield exacerbated by flooding on your Maryland farm, here's a structured plan combining immediate actions and arrangements for your absence:

**Immediate Actions (Before Traveling)**

1. **Drainage Improvement:**

   - **Create Ditches**: Dig shallow trenches around kale beds to divert excess water.
   - **Add Organic Matter**: Mix compost or aged manure into soil to improve structure (if soil isn't fully waterlogged).
   - **Sand Application**: Sprinkle coarse sand around plants to enhance drainage temporarily.

2. **Disease & Pest Control:**

   - **Fungicides**: Apply copper-based fungicides to combat root rot/downy mildew.
   - **Pesticides**: Use slug bait (iron phosphate) and insecticidal soap for aphids/slugs.
   - **Remove Affected Plants**: Prune damaged leaves and uproot severely infected plants.

3. **Soil Nutrition:**

   - **Balanced Fertilizer**: Apply a granular NPK (10-10-10) with calcium to replenish nutrients lost to flooding.
   - **Foliar Spray**: Use a liquid fertilizer (e.g., seaweed extract) for quick nutrient uptake.

4. **Weed & Moisture Management:**

   - **Mulching**: After water recedes, apply straw mulch to suppress weeds and retain soil moisture.

**During Your Absence**

1. **Delegate Monitoring:**

   - Arrange for a trusted person to check fields weekly for waterlogging, pests, and disease. Provide clear instructions for emergencies (e.g., reapplying fungicides).

2. **Slow-Release Fertilizer:**

   - Apply a slow-release granular fertilizer to sustain nutrient levels.

3. **Automate Drainage:**

   - Install temporary French drains or pump systems if flooding persists.

**Long-Term Strategies (Post-Travel)**

1. **Raised Beds**: Construct permanent raised beds to prevent future flooding.

2. **Soil Testing**: Conduct a pH/nutrient test to guide amendments (target pH 6.0–7.5).

3. **Resistant Varieties**: Plant flood-tolerant kale varieties (e.g., 'Winterbor' or 'Red Russian').

4. **Cover Crops**: Sow clover or rye post-harvest to improve soil health.

5. **Crop Rotation**: Rotate with non-brassicas (e.g., legumes) to break pest/disease cycles.

**Additional Tips**

- **Weather Monitoring**: Use apps like FarmWise to track rainfall and adjust plans.
- **Shade Cloth**: Deploy if heat stress follows flooding to protect plants.

By implementing these steps, you can mitigate immediate flooding impacts and establish resilience for future seasons. Safe travels!

**Expert Comment:** Pruning the damaged leaves is not a reasonable suggestion depending on the size of the farm. Same is true for adding sand around the plants. Model should ask the size of the farm.

---

### Error Pattern: *Misunderstanding crop lifecycles/regional fit*

**Example Question:** *What cover crop should I use in Alaska to increase biodiversity in my lettuce rotation?*

**Response**:
To enhance biodiversity in your Alaska lettuce rotation, consider a cover crop mix that addresses cold tolerance, quick growth, soil health, and ecological benefits. Here's a tailored recommendation:
**Cover Crop Recommendations:**

1. **Legumes for Nitrogen Fixation:**
   - **Alsike Clover (Trifolium hybridum)**: Cold-tolerant (Zones 3-8), perennial, fixes nitrogen, and supports pollinators when flowering. Plant in late summer to establish before frost; it may overwinter in milder areas.
   - **Field Peas (Pisum sativum)**: Quick-growing annual legume. Opt for cold-tolerant varieties like Arctic" or Austrian Winter" peas if planting early spring or late summer.

2. **Grasses for Biomass  Erosion Control:**
   - **Oats (Avena sativa)**: Fast-growing, winter-killed by frost, leaving residue that decomposes easily. Plant in late summer post-lettuce harvest.
   - **Barley (Hordeum vulgare)**: Slightly more cold-tolerant than oats; use for quicker biomass in short seasons.

3. **Brassicas for Biofumigation & Soil Structure:**
   - **Forage Radish (Raphanus sativus)**: Penetrates compacted soil, suppresses pests, and winter-kills. Ensure planting 6–8 weeks before frost for adequate growth.
   - **Mustard Greens (Sinapis alba)**: Quick-growing; releases compounds that suppress soil pathogens. Choose cold-hardy varieties.

4. **Pollinator Support:**
   - **Phacelia (Phacelia tanacetifolia)**: Cold-tolerant, fast-flowering annual that attracts beneficial insects. Blooms in 6–8 weeks, ideal for late summer planting.

**Sample Mix:**

- **"Alaska Biodiversity Blend"**: Combine oats (50
  - **Benefits**: Oats provide quick ground cover, clover fixes nitrogen and may overwinter, and phacelia supports pollinators.

**Planting Tips:**

- **Timing**: Sow immediately after lettuce harvest (late August/early September). Ensure seeds have 4–6 weeks before first frost.
- **Soil Prep**: Lightly till and ensure good seed-to-soil contact. Use a nurse crop (oats/barley) to protect slower-growing clover.
- **Termination**: Winter-killed crops (oats, radish) require no spring management. Mow/clip overwintered clover in spring before lettuce planting.

**Local Considerations:**

- Consult the **University of Alaska Fairbanks Cooperative Extension Service** for region-specific trials (e.g., interior vs. coastal climates).
- Test small plots first to gauge performance in your microclimate.

This mix enhances soil fertility, breaks pest cycles, and supports pollinators, aligning with Alaska's short growing season while boosting biodiversity.

**Expert Comment:** This makes the proposed approach unviable. Some suggestions, such as planting the cover crop in August, are not realistic. In Alaska, cover crops are typically planted from mid- to late July.

---

**Error Pattern:** *Irrelevant/Confusing response to question intent*

**Example Question:** *What steps should I take to recover my soybean field in Arkansas, which has been damaged by tornadoes and hail, given that I use conventional practices on a commercial farm with moderately shallow, high sulfur soil and easy access to chemicals?*

**Response**:
Recovering your soybean field in Arkansas after tornado and hail damage requires a structured approach tailored to your conventional practices, soil type, and access to resources. Here's a step-by-step plan:

1. **Immediate Damage Assessment**
   - **Crop Evaluation**:
     - Determine the percentage of plants severely damaged (broken stems vs. lodged plants). If ¿50
     - Check for shredded leaves, stem bruising, or root exposure due to erosion.
   - **Soil Inspection**:
     - Look for erosion, compaction, or debris. Test soil for pH, sulfur levels, and nutrients (N-P-K) to guide amendments.

2. **Replanting Decision**
   - **Timing**:
     - Arkansas's soybean planting window typically extends into June. If replanting by early June is feasible, use a maturity group suited to your remaining growing season.
     - If too late, explore short-season varieties or alternative crops (e.g., sorghum) if permitted by crop insurance.

3. **Soil Management**
   - **Amendments**:
     - Apply lime if soil pH is ¡6.0 (high sulfur may acidify soil). Balance sulfur with gypsum or other amendments if tests indicate excess.
     - Use conventional fertilizers (e.g., 10-10-10) based on soil test results, prioritizing phosphorus and potassium if roots are damaged.
   - **Erosion Control**:
     - Consider minimal tillage to preserve soil structure. If erosion is severe, add organic matter or silt fences to stabilize soil.

4. **Pest and Disease Control**
   - **Scouting**: Monitor for pathogens (e.g., *Phytophthora*, *Septoria*) and pests (e.g., bean leaf beetles).
   - **Chemical Applications**:
     - Apply fungicides (e.g., strobilurins) preventively if hail caused plant wounds.
     - Use herbicides (e.g., glyphosate for burndown, post-emergents like clethodim) to control weeds in damaged areas.

5. **Irrigation and Drainage**

- Ensure fields are well-drained to prevent waterlogging. Adjust irrigation schedules to avoid drought stress, especially in shallow soils.

6. **Crop Recovery Support**

- **Foliar Feeding**: Apply a balanced foliar fertilizer (micronutrients + low N) to support stressed plants.
- **Nitrogen Boost**: If root nodules are damaged, a small N application (20–30 lbs/acre) may aid recovery.

7. **Infrastructure Repair**

- Clear debris to prevent equipment damage. Repair irrigation systems, storage units, or fences impacted by the tornado.

8. **Documentation and Insurance**

- Photograph damage and maintain records for insurance claims. Contact your provider promptly to discuss replanting compensation or loss coverage.

9. **Long-Term Risk Mitigation**

- **Cover Crops**: Plant winter cover crops (e.g., rye) post-harvest to reduce future erosion.
- **Diversification**: Consider crop rotation or insurance add-ons for extreme weather resilience.

**Key Considerations:**

- **Conventional Flexibility**: Leverage chemical access for rapid response but avoid overapplication to prevent soil/plant stress.
- **High Sulfur Soil**: Monitor for pH shifts and nutrient lockout; adjust amendments as needed.

By following these steps, you can optimize recovery while maintaining soil health and productivity for future seasons.

**Expert Comment:** A lot of things that were included in the answer were not necessary to the answering of the original question.

---

**Error Pattern:** *Incorrect pest/weed behavior attribution*

**Example Question:** *What parts of wheat are affected by the dirt-colored seed bug in Wyoming?*

**Response**:
The dirt-colored seed bug (likely in the family Lygaeidae or Rhyparochromidae) primarily affects the **seeds (kernels)** of wheat plants in Wyoming. Here's a detailed breakdown:

1. **Seeds (Kernels):**

- **Primary Target**: These bugs use piercing-sucking mouthparts to feed on developing or mature seeds within the wheat head. This occurs during the milky or dough stages of grain development.
- **Damage**: Feeding punctures can cause shriveled, discolored, or deformed seeds, reducing yield and grain quality. Infested seeds may also be more susceptible to fungal infections or fail to germinate.

2. **Potential Secondary Effects:**

- **Early-Season Feeding**: Nymphs or adults might occasionally feed on softer plant tissues (e.g., stems, leaves) if seeds are unavailable, though this is less common.
- **Indirect Impact**: Heavy infestations can lead to significant economic losses due to compromised seed viability and marketability.

3. **Timing:**

- Activity peaks during wheat's reproductive stages (heading to maturity), aligning with seed availability in Wyoming's growing season.

Farmers should monitor wheat heads for bugs and damaged kernels, particularly during grain fill. Management strategies may include timely insecticide applications or crop rotation to mitigate infestations.

**Expert Comment:** The dirt colored seed bug may eat fallen seeds in a field, but is not an important economic pest for wheat.

---

**Error Pattern:** *Outdated or banned chemical advice*

**Example Question:** *How can I prevent Wild safflower from growing in my lettuce field next year, given that I live in California, use conventional farming practices, and operate a small farm without hail for a long time, with my kids helping out?*

**Response**:
To prevent Wild safflower from growing in your California lettuce field next year, consider integrating the following conventional strategies, tailored to your small farm and family involvement:

1. **Pre-Emergent Herbicides**

- **Selective Herbicides**: Apply pre-emergent herbicides labeled for lettuce that target Wild safflower. Examples include **Pendimethalin** or **DCPA (Dacthal)**, which inhibit weed germination. Consult local agricultural extensions for California-specific recommendations.
- **Timing**: Apply before Wild safflower germination, typically in early spring or fall, depending on its lifecycle.

2. **Cultural Practices**

- **Crop Rotation**: Rotate with crops like broccoli or onions, which compete better with weeds or have different herbicide options to reduce Wild safflower seed banks.
- **Stale Seedbed Technique**:
  (a) Prepare the seedbed 2–3 weeks before planting lettuce.
  (b) Irrigate to trigger weed germination.
  (c) Use shallow tillage or a non-residual herbicide (e.g., **glyphosate**) to kill emerged weeds before planting lettuce.

3. **Mechanical Control**

- **Tillage**: Shallow tillage during Wild safflower's germination phase (often early spring) to disrupt seedlings. Avoid deep plowing, which may bring buried seeds to the surface.
- **Hand Weeding**: Engage your kids in regular, early-season hand-weeding to remove Wild safflower before it sets seed. Focus on removing plants pre-flowering to prevent seed dispersal.

4. **Soil Solarization**

- **Process**: During hot summer months (June–August), cover moist soil with clear plastic for 4–6 weeks. Solar heat kills weed seeds and pathogens.
- **Effectiveness**: Highly effective in California's climate and reduces reliance on chemicals.

5. **Sanitation and Field Hygiene**

- **Equipment Cleanliness**: Clean tools and machinery to avoid spreading seeds.
- **Edge Management**: Mow or herbicide field borders to prevent Wild safflower from encroaching.

6. **Mulching**

- **Plastic Mulch**: Use black or clear plastic mulch to suppress weeds. Ideal for lettuce rows, as it warms soil and blocks light to weed seeds.

7. **Monitoring and Timing**

- **Scout Regularly**: Inspect the field weekly for early Wild safflower growth. Early detection simplifies control.
- **Adjust Planting Dates**: Plant lettuce earlier or later to avoid peak Wild safflower germination periods.

8. **Dense Planting**

- **Optimize Spacing**: Plant lettuce at recommended dense spacing to shade out weeds without compromising head size.

**Safety and Family Involvement**

- **Herbicide Safety**: Choose herbicides with low toxicity and follow re-entry intervals (REIs) to ensure safety for your kids. Use protective gear during application.

- **Educational Opportunity**: Teach your children weed identification and sustainable farming practices through hands-on weeding and monitoring.

By combining these methods, you can effectively suppress Wild safflower while maintaining a productive, family-friendly lettuce operation.

**Expert Comment:** However, from the latest sources from Google, DCPA is banned for use due to health concerns.

## F  LLM-BASED RESPONSE FILTERING PROMPT (AGTHOUGHTS)

The following prompt was used with GPT-4.1 to evaluate the correctness of responses in AGTHOUGHTS. It guides the evaluation based on five predefined criteria and incorporates common error types that are likely to appear in the dataset. These error trends were analyzed through expert comments, as discussed in section E.

```
############################################################
# Prompt Used for LLM Filtering to Finalize AgThoughts #
############################################################

You are an expert agronomist and evaluator.
Your job is to judge the quality of agricultural answers provided to
    farmers,
based on domain-specific knowledge, scientific accuracy,
and practical context.

You will be given:
- A question from a user
- A proposed answer
Your task is to rigorously evaluate the answer using the rubric below
and domain-specific insights that reflect common issues found in
    agronomic advice.

---

### RUBRIC FOR EVALUATION

Judge the answer based on the following five dimensions:

1. **Factual Accuracy**
   - Does the answer avoid biological or chemical inaccuracies?
   - Is pest/crop behavior and soil-chemistry accurately represented?

2. **Contextual Relevance**
```

```
    - Is the advice regionally and seasonally appropriate?
    - Does it consider the specific crop and geographic location?
    - Does it avoid over-generalization?

3. **Practical Feasibility**
    - Is the recommendation realistic given likely labor, cost, or scale
       constraints?

4. **Logical Consistency**
    - Does the answer contradict itself?
    - Are the steps or claims internally coherent?

5. **Completeness**
    - Does it request needed clarifying info (e.g., soil test, crop stage)?
    - Are critical details omitted?

---

### DOMAIN INSIGHTS (COMMON ERRORS TO WATCH FOR)

You must watch out for these common patterns of poor answers:
- **Unverified or inaccurate crop recommendations** (e.g., suggesting
   tomato varieties unsuited to the region)
- **Generic fertilizer/pesticide advice** without a soil test or label
   check
- **Oversimplified lifecycle assumptions**, like planting or harvest
   timings that ignore local climate
- **Ignoring feasibility**, e.g., hand-pruning corn, applying sand around
    each kale plant
- **Wrong attribution**, e.g., pests misidentified or overstated
- **Contradictions**, like calling a pest harmless but recommending
   treatment
- **Recommendations for banned or unlabeled chemicals**
- **Treating complex issues as single-cause problems** without linking
   factors
- **Wrong stage/context addressed**, e.g., giving post-harvest tips to a
   pre-harvest question

---

### YOUR OUTPUT FORMAT

Respond with the following format:

{{
"Factual Accuracy": "Pass / Needs Improvement / Fail",
"Contextual Relevance": "Pass / Needs Improvement / Fail",
"Practical Feasibility": "Pass / Needs Improvement / Fail",
"Logical Consistency": "Pass / Needs Improvement / Fail",
"Completeness": "Pass / Needs Improvement / Fail",
"Matched Error Patterns": [
  "e.g., Generic fertilizer advice without context",
  "e.g., Recommending unverified variety for the region"
],
"Most Critical Flaw & Fix": "Describe the main issue in 1-2 sentences."
}}

---

Now, evaluate the following:

**Question:**
{}

**Answer:**
```

```
{}
```

## G  RUBRIC FOR SCORE CALCULATION AND VERDICT

The AGTHOUGHTS LLM Filter evaluates responses based on the following criteria: Factual Accuracy, Contextual Relevance, Practical Feasibility, Logical Consistency, and Completeness. Each response is assigned a decision—pass, needs improvement, or fail—for each criterion. A pass is scored as 2 points, needs improvement as 1 point, and fail as 0 points. The criteria are weighted unevenly to emphasize the relative importance of some over others. The rubric used to compute the evaluation score for a response is outlined in Table 6. Based on the scores computed using the rubric, we apply a threshold to filter high-quality responses, as outlined in Table 7.

Table 6: Evaluation Rubric Dimensions, Weights, and Scoring Scale

| Dimension | Weight | Pass (2) | Needs Improvement (1) | Fail (0) |
|---|---|---|---|---|
| Factual Accuracy | 3 | 6 | 3 | 0 |
| Contextual Relevance | 2 | 4 | 2 | 0 |
| Practical Feasibility | 1 | 2 | 1 | 0 |
| Logical Consistency | 2 | 4 | 2 | 0 |
| Completeness | 1 | 2 | 1 | 0 |
| **Total Possible Score** | – | **18** | – | – |

Table 7: Final Verdict Logic Based on Evaluation Outcome

| Condition | Verdict |
|---|---|
| Factual Accuracy is `Fail` OR more than 2 dimensions are `Fail` | Reject |
| Total score $\geq 17$ | High Quality |
| Total score $< 17$ | Reject |

## H  SCORE DISTRIBUTION OF EVALUATED RESPONSES

We compute the distribution of scores across the entire dataset and present the log-scale distribution in figure 10. Approximately 13.5% of the responses received a score below 17; these were excluded from further analysis. The remaining responses were retained.

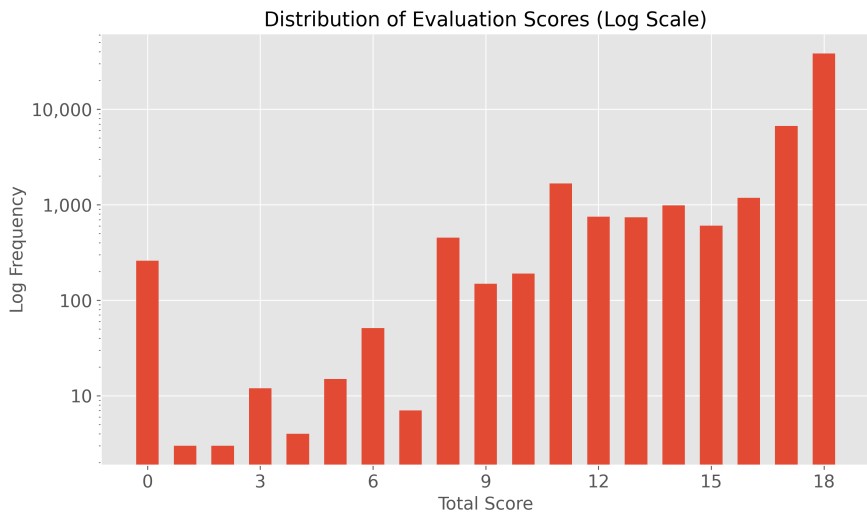

Figure 9: Distribution of Evaluation Scores (Log Scale). The x-axis shows the total score (max = 18), and the y-axis shows the log-scaled frequency of responses receiving that score.

## I    EVOLUTION OF AGREASON FROM AGTHOUGHTS

A set of 100 question–answer pairs from AGTHOUGHTS was curated to generate high-quality responses through a two-stage refinement process. First, domain experts manually filtered the original AGTHOUGHTS responses to retain only the most informative content (see I for examples). Second, these cropped segments were further refined using a large language model (LLM) to ensure semantic consistency and clarity (see I for examples).

**AGTHOUGHTS Question**: How can I prevent low-grade potato harvest next year given that I live in Missouri and currently experience low harvest quality?

**AGTHOUGHTS Response**:

> To prevent a low-grade potato harvest in Missouri next year, address the following key areas systematically:
> **1. Soil Health**
> **Test Soil**: Conduct a soil test to determine pH (ideal: 5.0–6.5) and nutrient levels. Amend with lime (to raise pH) or sulfur (to lower pH) as needed.
> **Enrich Soil**: Incorporate organic matter (compost, well-rotted manure) to improve drainage, structure, and fertility. Focus on potassium and phosphorus for tuber development.
> **2. Crop Rotation**
> **Rotate Crops**: Avoid planting potatoes in the same plot for 3–4 years. Rotate with legumes (e.g., beans, peas) to fix nitrogen and break pest/disease cycles.
> **3. Disease Management**
> **Resistant Varieties**: Choose disease-resistant cultivars (e.g., 'Kennebec' for blight, 'Russet' for scab).
> **Sanitation**: Remove plant debris post-harvest to reduce disease carryover.
> **Watering**: Use drip irrigation to keep foliage dry and prevent fungal diseases. Water consistently, especially during tuber formation.
> **4. Pest Control**
> **Monitoring**: Regularly inspect for Colorado potato beetles, wireworms, and aphids.
> **IPM Strategies**: Use row covers, handpick pests, apply neem oil or spinosad, and encourage beneficial insects (e.g., ladybugs).
> **5. Planting Practices**

**Timing**: Plant after the last frost when soil reaches 45–50°F (typically March–April in Missouri). Avoid waterlogged soil.

**Spacing**: Plant seed potatoes 12–15 inches apart in rows 30–36 inches apart to improve airflow and reduce disease.

**6. Water and Nutrient Management**

**Consistent Moisture**: Maintain even soil moisture; mulch with straw to retain water and suppress weeds.

**Balanced Fertilization**: Use a low-nitrogen, high-phosphorus/potassium fertilizer (e.g., 5-10-10) to prioritize tuber growth over foliage.

**7. Hilling and Mulching**

**Hill Plants**: Mound soil around stems when plants are 6–8 inches tall to protect tubers from sunlight (preventing greening) and improve yield.

**Mulch**: Apply organic mulch to regulate soil temperature and moisture.

**8. Harvest and Storage**

**Timely Harvest**: Dig potatoes when vines die back (usually 2–3 weeks after flowering). Avoid harvesting in wet conditions.

**Cure and Store**: Cure tubers at 50–60°F with high humidity for 10–14 days, then store in a cool, dark place (38–40°F).

**9. Climate Adaptation**

**Heat Tolerance**: Select short-season varieties (e.g., 'Yukon Gold') to avoid peak summer heat. Use shade cloth if extreme heat is forecasted.

**10. Record-Keeping**

Track planting dates, treatments, and weather patterns to refine strategies annually.

By systematically addressing these factors, you can optimize soil conditions, mitigate pests/diseases, and improve overall potato quality and yield in Missouri's climate. Start with a soil test and plan rotations early for the best results.

---

**AGTHOUGHTS response after expert filtering**:

**1. Soil Health**

**Test Soil**: Conduct a soil test to determine pH (ideal: 5.0–6.5) and nutrient levels. Amend with lime (to raise pH) or sulfur (to lower pH) as needed.

**Enrich Soil**: Incorporate organic matter (compost, well-rotted manure) to improve drainage, structure, and fertility. Focus on potassium and phosphorus for tuber development.

**2. Crop Rotation**

**Rotate Crops**: Avoid planting potatoes in the same plot for 3–4 years. Rotate with legumes (e.g., beans, peas) to fix nitrogen and break pest/disease cycles.

**3. Disease Management**

**Resistant Varieties**: Choose disease-resistant cultivars (e.g., 'Kennebec' for blight, 'Russet' for scab).

**Sanitation**: Remove plant debris post-harvest to reduce disease carryover.

**5. Planting Practices**

**Timing**: Plant after the last frost when soil reaches 45–50°F (typically March–April in Missouri). Avoid waterlogged soil.

**Spacing**: Plant seed potatoes 12–15 inches apart in rows 30–36 inches apart to improve airflow and reduce disease.

**Balanced Fertilization**: Use a low-nitrogen, high-phosphorus/potassium fertilizer (e.g., 5-10-10) to prioritize tuber growth over foliage.

**Timely Harvest**: Dig potatoes when vines die back (usually 2–3 weeks after flowering). Avoid harvesting in wet conditions.

**Cure and Store**: Cure tubers at 50–60°F with high humidity for 10–14 days, then store in a cool, dark place (38–40°F).

---

**Processed AGREASON response**:

To prevent a low-grade potato harvest next year in Missouri, first, conduct a soil test to check pH and nutrient levels, adjusting with lime or sulfur as needed. Enhance soil by adding organic matter like compost or well-rotted manure, focusing on potassium and phosphorus. Implement crop rotation by avoiding planting potatoes in the same plot for 3-4 years and rotate with legumes. Select disease-resistant potato varieties and remove plant debris after harvest. Plant after the last frost when soil is 45–50°F, ensuring proper spacing of 12–15 inches apart in rows 30–36 inches apart. Use a low-nitrogen, high-phosphorus/potassium fertilizer. Harvest when vines die back and avoid wet conditions. Cure tubers at 50–60°F with high humidity for 10–14 days, then store them in a cool, dark place at 38–40°F.

## J AGREASON BENCHMARK EXAMPLES

**Crop Management**

**Question:** *What is the most cost-effective method to fertilize crops on moderately shallow soil, given that we had early-season sprinkles?*

**Answer:** To address fertilization on moderately shallow soil after early-season sprinkles, combine grid soil testing, starter banding of 20–30 lb $P_2O_5$/A and 10–15 lb $N$/A polymer-coated urea, variable-rate split N (40 % planting, 30 % V6, 30 % V12) guided by optical sensing, NBPT and DCD inhibitors, zone-banded composted manure (0.5–1 ton/A) plus biochar (1 ton/A), legume cover-crop N credits, and co-op-shared VRT equipment to minimize cost while maintaining yield.

**Biotic Diseases**

**Question:** *How can I prevent Wild safflower from growing in my lettuce rows on a small farm without hail for a long time, with my kids helping out?*

**Answer:** Uproot existing safflower and sanitize tools; incorporate trifluralin (0.75 lb ai/A) pre-plant; wheel-hoe, flame-weed, or finger-weed seedlings; mulch or lay woven fabric between rows; rotate with cereals and allelopathic rye; trial *Pseudomonas* seed treatments and mycorrhizae; create kid-friendly scouting games; and scout weekly with photo records for rapid response.

**Abiotic Soil**

**Question:** *How can I manage moderately shallow soil with low iron content for a barley crop in Utah using conventional practices on a commercial farm with high-tech equipment, given the prolonged lack of cloud cover?*

**Answer:** GPS-guided subsoiling, variable-rate Fe-EDDHA (6–8 lb Fe/A) plus foliar Fe, in-furrow sulfur to lower pH, capacitance-probe-scheduled deficit irrigation, shallow-soil barley cultivars, VRT top-dress N and K from drone NDVI maps, windbreaks, cover-crop rye, weekly multispectral drones for chlorosis, and annual compost injection restore yield on shallow, iron-poor soil under high insolation.

**Abiotic Harvest**

**Question:** *What methods can I use to prevent a low harvest of squash in Vermont?*

**Answer:** Construct 8–12 in. raised beds, apply black plastic mulch, transplant early-maturing cultivars under row covers, incorporate balanced compost and side-dress N at bloom, irrigate 1 in./week via drip, plant bee strips and hand-pollinate if cloudy, apply potassium bicarbonate or sulfur at first mildew sign, use covers on cold nights and shade cloth on hot days, and harvest every 2–3 days to maintain high yields.

## Abiotic Weather

**Question:** *How can I save my mustard crop from ice damage in California?*

**Answer:** Plant cold-tolerant cultivars, set frost alerts at $38°F$, deploy row covers or Wall-of-Water, run propane heaters and wind machines on freeze nights, apply overhead sprinkler icing at 0.05 in./hr, continue until sunrise, then foliar-apply K and Ca for recovery and use cover crop stubble to insulate soil.

## Abiotic Nutrients

**Question:** *What is affecting my tangerines on my commercial organic farm in Florida, given early-season wind and my lack of experience with this situation?*

**Answer:** Wind scald, salt spray, and wind-induced Ca/B deficiencies are common—install windbreaks to halve wind speed, rinse foliage weekly, foliar-apply 4 lb $CaCl_2$/A + 2 lb borax/A twice 10 days apart, check compost C:N (20–30:1), incorporate sunn-hemp residues 30 days pre-plant, cover exposed roots with compost + mulch, ensure drip delivers 1 in./week, and release predatory mites to counter stress-induced pests.

## Biotic Insects

**Question:** *How can I use pheromone traps to scout for insects in mustard crops in Kentucky?*

**Answer:** Hang sticky delta traps at canopy height (1 trap/2 acres) baited for diamondback moth and cabbage looper, check twice weekly, start Bt or spinosad sprays when weekly catch exceeds 50 moths, log GPS counts, calibrate trap:field ratios bi-weekly, and intensify to 1 trap/acre during seedling stage.

## Biotic Weeds

**Question:** *What chemicals can I use to manage weeds in my potato crop in Montana?*

**Answer:** Apply pendimethalin (2.1 qt/A) and S-metolachlor (1.5 pt/A) pre-plant, metribuzin (0.5–1 pt/A) + ethofumesate (1.5 pt/A) early post, clopyralid + fluroxypyr (1.5 qt/A) mid-season, and clethodim (12–16 oz/A) or carfentrazone (2–4 oz/A) late; rotate HRAC groups and cultivate between sprays.

## Cover Crop

**Question:** *What cover crop should I grow alongside cotton in Tennessee that has recently been cold, and I planted on my insurance date?*

**Answer:** Drill cereal rye at 60 lb/A immediately after cotton; it germinates in cool soils, builds biomass, buffers temperature by 5–8 °F, suppresses weeds, and can be roller-crimp terminated at boot stage before the next crop.

## Management

**Question:** *What steps should I take to care for my sunflowers in Delaware?*

**Answer:** Ensure $\geq 8$ h sun, adjust pH to 6.0–7.5 with compost, plant 2 weeks after last frost at 12–18 in. spacing, band 10-10-10 (1 lb/100 $ft^2$), side-dress 1 lb N/A at V4–V6, irrigate 1–1.5 in./week, mulch 2–3 in. scout for sunflower moth and wilt, stake varieties > 6 ft, and harvest when back of head turns tan.

## K    STATISTICS OF AGREASON

figure 10 presents the distribution of key attributes in the AGREASON dataset. We analyze the composition of the dataset across several dimensions, including question category, geographic location, crop type, farming practice, and farm size. This breakdown highlights the diversity and coverage of AGREASON.

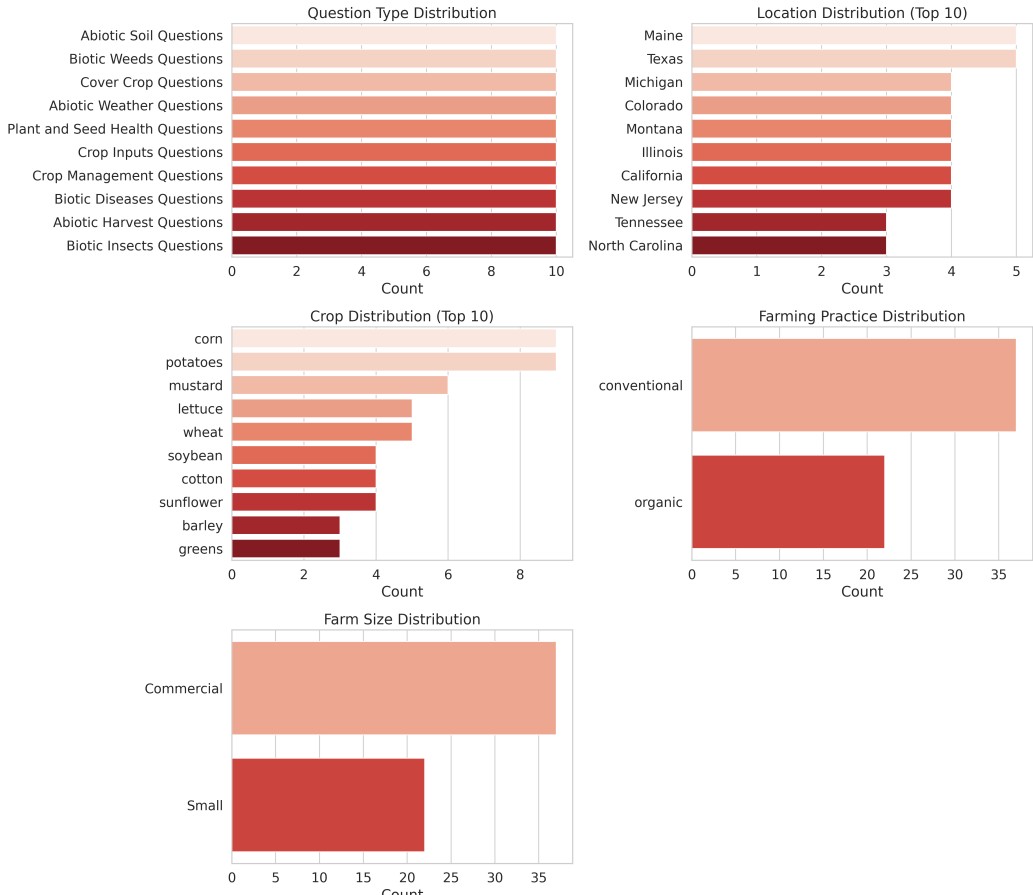

Figure 10: Distribution of question attributes in AGREASON, including location, crop type, and modifiers such as farm size and practices.

# L AGREASON LLM-AS-A-JUDGE PROMPT TEMPLATES

Listing 1: LLM-as-a-Judge Prompt for Benchmarking

```
##################################################
# LLM-as-a-Judge Prompt for Benchmarking #
##################################################

You are a neutral evaluator.
You will receive:
 a **User Query**
 a **Ground-Truth Answer** (reference)
 a **Model-Generated Response**

**Goals**
1. Judge each *verifiable factual statement* in the model's response
    against the ground-truth answer.
2. List every ground-truth fact that the model failed to mention.

TASK

### Part A - Label response statements
1. Break the model response into standalone factual statements (skip
    headings, greetings, fluff, or nonverifiable text).
2. For **each** statement, output one JSON line with:
    "sentence" - the factual statement
    "label" - one of "supported", "unsupported", "contradictory"
    "rationale" - brief justification (1-2 sentences)
    "excerpt" - supporting / contradicting text from the groundtruth, or "
       none" for unsupported

### Part B - List missing groundtruth facts
3. Break the groundtruth answer into discrete facts (one per line).
4. Identify which of those facts were **not** covered in PartA.
5. For every uncovered fact, output one JSON line with:
    "missing_fact" - the fact text
    "note" - always "not covered by the model response"

OUTPUT FORMAT

# Block 1  labeled response statements
{"sentence": "...", "label": "...", "rationale": "...", "excerpt": "..."}

# Block 2  missing groundtruth facts
{"missing_fact": "...", "note": "not covered by the model response"}

*Only output lines for statements you labeled (Block 1) and for unmatched
    groundtruth facts (Block 2).*
```

Listing 2: Per-Example Input Given to the Judge Model

```
INPUT

User Query:
{query}

GroundTruth Answer:
{ground_truth}

Model Response:
{model_response}
```

## M    EVALUATION METRICS

$$\text{Precision} = \frac{\text{TP}}{\text{TP} + \text{FP}_u + \text{FP}_c}, \tag{1}$$

$$\text{Recall} = \frac{\text{TP}}{\text{TP} + \text{FN}}, \tag{2}$$

$$\text{F1-score} = 2 \cdot \frac{\text{Precision} \times \text{Recall}}{\text{Precision} + \text{Recall}}. \tag{3}$$

## N    EVALUATION OF REASONING MODELS ON AGREASON

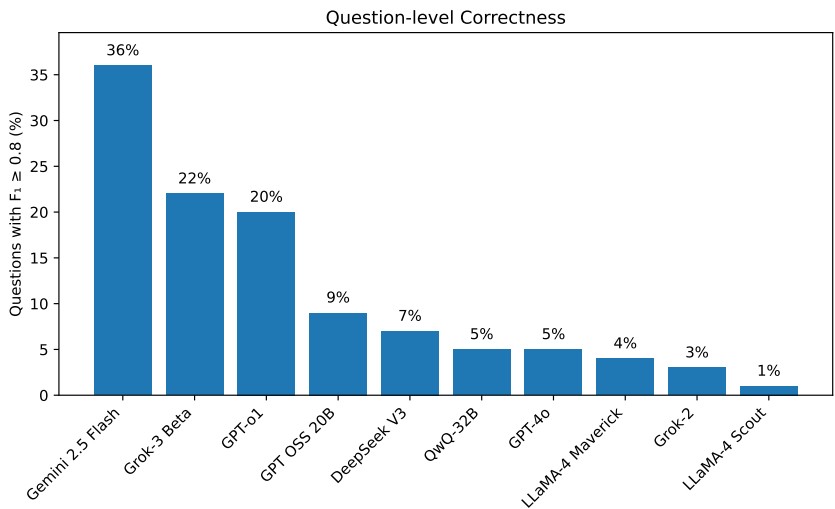

Figure 11: Question-level pass rate ($F_1 \geq 0.8$) for the base models.

| Model | Score ($F_1 > 0.80$) | Precision | Recall | TP | FP | FN |
|---|---|---|---|---|---|---|
| Gemini 2.5 Flash | 36.0% | 0.727 | 0.778 | 1797 | 676 | 514 |
| Grok-3 Beta | 22.0% | 0.583 | 0.815 | 2024 | 1450 | 459 |
| GPT-o1 | 20.0% | 0.654 | 0.710 | 1343 | 711 | 549 |
| Phi-3 14B SFT | 13.0% | 0.564 | 0.719 | 1085 | 840 | 424 |
| Qwen2.5 14B Full SFT | 9.0% | 0.560 | 0.681 | 988 | 776 | 462 |
| GPT OSS 20B | 9.0% | 0.534 | 0.731 | 1390 | 1213 | 512 |
| Mistral 7B Full SFT | 7.0% | 0.526 | 0.628 | 944 | 849 | 558 |
| DeepSeek V3 | 7.0% | 0.544 | 0.644 | 1037 | 868 | 574 |
| Phi-3 3B Full SFT | 7.0% | 0.524 | 0.661 | 993 | 901 | 509 |
| Phi-3 3b SFT | 5.0% | 0.474 | 0.598 | 947 | 1051 | 636 |
| Qwen2.5 3B Full SFT | 5.0% | 0.514 | 0.658 | 984 | 929 | 511 |
| QwQ-32B | 5.0% | 0.505 | 0.693 | 1244 | 1219 | 552 |
| GPT-4o | 5.0% | 0.554 | 0.558 | 821 | 660 | 650 |
| LLaMA-3 3B Full SFT | 4.0% | 0.518 | 0.622 | 947 | 880 | 576 |
| LLaMA-4 Maverick | 4.0% | 0.596 | 0.593 | 1014 | 688 | 696 |
| Mistral 7B SFT | 3.0% | 0.470 | 0.678 | 1232 | 1389 | 586 |
| Qwen2.5 8B Full SFT | 3.0% | 0.503 | 0.644 | 950 | 938 | 525 |
| Qwen2.5 14B | 3.0% | 0.515 | 0.533 | 811 | 763 | 710 |
| LLaMA-3 3B SFT | 3.0% | 0.372 | 0.399 | 530 | 896 | 797 |
| Grok-2 | 3.0% | 0.466 | 0.575 | 883 | 1010 | 652 |
| Qwen2.5 3B | 1.0% | 0.422 | 0.501 | 728 | 996 | 725 |
| LLaMA-4 Scout | 1.0% | 0.480 | 0.523 | 798 | 863 | 729 |
| Phi-3 14B | 1.0% | 0.548 | 0.400 | 517 | 427 | 774 |
| Qwen2.5 8B | 0.0% | 0.457 | 0.513 | 762 | 907 | 722 |
| Phi-3 3B | 0.0% | 0.440 | 0.452 | 614 | 780 | 745 |
| Mistral 24B | 0.0% | 0.442 | 0.557 | 834 | 1054 | 663 |
| LLaMA-3 8B | 0.0% | 0.183 | 0.088 | 86 | 384 | 895 |
| Mistral 7B | 0.0% | 0.408 | 0.520 | 792 | 1150 | 731 |

Table 8: Per-model benchmark performance

## O  COMPUTE RESOURCES

We utilize A100 GPU from Iowa State University's High Performance Computing Cluster for our Supervised Fine-Tuning experiments. For inference with proprietary models such as GPT, Gemini, and Claude, we access their respective APIs. Additionally, we use the Together.ai API to perform inference on selected open-source models included in our evaluations.

