# OpenReview forum: "Towards Large Reasoning Models for Agriculture"
_ICLR.cc/2026/Conference — ICLR 2026 Conference Withdrawn Submission_

### Official Review · Reviewer_r91M · 2025-10-15

**Soundness:** 2
**Presentation:** 2
**Contribution:** 2
**Rating:** 4
**Confidence:** 3

**Summary:**

This paper addresses the complex reasoning requirements of agricultural decision-making by proposing Large Reasoning Models (LRMs) and releasing three core resources: (1) AGREASON — the first expert-curated, open agricultural reasoning benchmark (100 questions); the strongest baseline (Gemini) achieves only 36% accuracy. (2) AGTHOUGHTS — a 44.6K QA dataset with explicit reasoning traces, covering 10 agricultural topic categories. (3) AGTHINKER — a lightweight reasoning model fine-tuned on AGTHOUGHTS that outperforms comparable open-source models on consumer-grade GPUs (e.g., Phi‑3 14B attains 13% accuracy after fine-tuning). Empirical results show LRMs substantially outperform conventional LLMs while leaving considerable room for improvement, underscoring the need for domain-specific reasoning models in agriculture.

**Strengths:**

1. Introduces AGREASON, the first expert-curated open-ended benchmark for agricultural reasoning (100 questions), addressing the lack of domain-specific evaluation tools.

2. Provides AGTHOUGHTS, a large-scale dataset (44.6K QA pairs) with synthetic reasoning traces, validated by agronomy experts, enabling fine-grained model training and evaluation.

3. Demonstrates the effectiveness of domain-specific fine-tuning: The AGTHINKER models (e.g., Phi-3 14B) achieve 13% accuracy on AGREASON, outperforming base models and some open-source alternatives.

4. Evaluates 18 models (open-source and proprietary) across 10 agronomic categories, revealing performance disparities (e.g., strong results in Biotic Diseases but weaknesses in Cover Crops).

**Weaknesses:**

1. AGREASON’s 100-question benchmark, while aligned with similar works (e.g., GPQA), may lack statistical power for robust generalization. Questions primarily cover U.S. states, limiting applicability to global agricultural contexts.

2. Human review sampled only 200 QA pairs (0.45% of the dataset), potentially overlooking systemic errors.

3. The manuscript does not convincingly demonstrate novelty, either in its data synthesis methods or in the benchmark design.

**Questions:**

See weakness.

---

### Official Review · Reviewer_QmYp · 2025-10-26

**Soundness:** 3
**Presentation:** 3
**Contribution:** 2
**Rating:** 4
**Confidence:** 3

**Summary:**

The paper introduces an open-ended reasoning benchmark and a suite of reasoning models for the domain of agriculture.

**Strengths:**

How to apply Large Reasoning Models (LRMs) to the domain of agriculture seems to be a research question that has the potential to make significant economic impacts but has not yet received much attention; the open-endedness of the proposed benchmark seems an improvement compared to existing ones.

**Weaknesses:**

While the datasets and models introduced by the paper are valuable for the intended community, the paper makes few technical contributions otherwise and may not be of interest to the general community. That said, I fully support publication of the paper in a relevant workshop at ICLR, if any.

**Questions:**

What are the unique challenges of applying LRMs to the domain of agriculture, if any?

---

### Official Review · Reviewer_nELz · 2025-10-29

**Soundness:** 2
**Presentation:** 3
**Contribution:** 3
**Rating:** 6
**Confidence:** 3

**Summary:**

This paper studies the large language model reasoning for agricultural problems. The contribution consists of two parts, including creating two datasets for training and testing, as well as training a small scale language model based on the constructed dataset.

For creating the dataset, the authors first define some templates, e.g. 'what do I do if my Y has X change', then generate concrete question-answer pairs based on the templates using LLM. Human experts then participate to spot problems in the generated data. This process is repeated to generate enough data.

**Strengths:**

This paper targets an important question. Guidance in agriculture, especially context-aware analysis and solutions customized to specific situations, is very crucial.

The generated dataset not only contain testing datasets to benchmark existing models, but also contain a large-scale training set contain CoT traces.

**Weaknesses:**

The generation process does not seem very valid. Specifically, the modifiers generate random instantiations independently, therefore very likely to result in invalid combinations, like the example shown in Figure 1. It is unclear how these invalid questions are filtered, either by human or by LLM. The introduced human validation is only applied to part of data sampled from the entire set. Although refinement is performed based on the validation, it is unclear whether the quality of the resulting datasets is good enough.

More details could be provided on how human experts perform the qualify check and refinement.

**Questions:**

1. Why not use part of the AGTHOUGHTS to enrich the testing set? Is it limited by the cost to enquiry the closed source models?

2. Are the AGTHOUGHTS and AFREASON generated in exactly the same way, since both are claimed to be supported by experts?

3. It would be better to show how human experts perform on these questions. Since most of reviewers may not have enough agricultural knowledge, it is hard to tell the low performance from the models results from lack of capacity or ill-defined questions.

4. Regarding the same question, is it possible to have different correct solutions? If so, then the evaluation may

---

### Official Review · Reviewer_b8WN · 2025-10-30

**Soundness:** 3
**Presentation:** 2
**Contribution:** 2
**Rating:** 4
**Confidence:** 4

**Summary:**

This paper introduces a suite of resources aimed at advancing the reasoning capabilities of large language models in the domain of agriculture. The authors argue that agricultural decision-making is highly contextual (geographic, climatic, etc.) and that existing benchmarks are insufficient for evaluating this nuanced reasoning. The main contributions are threefold:
1. AGREASON: An expert-curated, open-ended benchmark consisting of 100 challenging agricultural reasoning questions with gold-standard answers.
2. AGTHOUGHTS: A large-scale dataset of 44.6K question-answer pairs, where answers are augmented with synthetically generated reasoning traces.
3. AGTHINKER: A collection of smaller language models fine-tuned on the AGTHOUGHTS dataset, designed to be efficient while possessing domain-specific reasoning skills.

**Strengths:**

1. The AGREASON benchmark is a key strength. Its 100 questions were carefully curated and refined by domain experts, ensuring a high-quality, challenging evaluation.
2. This work evaluates a wide range of 18 recent open-source and proprietary models, providing a broad and up-to-date snapshot of current capabilities in this domain. The detailed, statement-level evaluation is far more insightful than simple metrics like ROUGE.
3. The authors demonstrate the value of AGTHOUGHTS by fine-tuning the AGTHINKER models and showing performance gains over the base models, validating that AGTHOUGHTS effectively transfers domain-specific reasoning abilities.

**Weaknesses:**

1. The quality of the AGTHOUGHTS dataset hinges on a GPT-4.1-based filter that was designed based on expert feedback from only 200 examples. There is no quantitative analysis of this filter's accuracy (e.g., its agreement with human experts). Without this, it is difficult to gauge the level of noise, error, or bias that may have been introduced into the training data, potentially limiting the ultimate performance of models trained on it.
2. The evaluation relies solely on an LLM-as-Judge for open-ended, non-verifiable answers. This methodology is susceptible to the judge model's inherent biases (e.g., verbosity, style). While the AGREASON benchmark is finely curated by human experts, its reliability could be further strengthened by including verifiable questions (e.g., multiple-choice questions).
3. The AGTHINKER models are only evaluated on the in-domain AGREASON benchmark. To more robustly claim the effectiveness and generalizability of the AGTHOUGHTS dataset, evaluation on other agricultural or scientific benchmarks is necessary.

**Questions:**

1. Could the authors provide more details of the LLM-based filtering pipeline used to curate the AGTHOUGHTS dataset? Was the filter's performance (e.g., accuracy, agreement with humans) evaluated on a hold-out set of human-annotated samples?
2. In lines 229-230, you mention that evaluators "revisited a subset of 100 high-quality responses to extract essential content elements" to form the AGREASON benchmark. Could you elaborate on this process? How were these "essential content elements" extracted?
3. In line 271, you mention that the judge model "processes the candidate response by decomposing it into individual statements". Could you elaborate in detail how this process is achieved?
4. The AGTHINKER models, fine-tuned on AGTHOUGHTS, were only evaluated on AGREASON. To better demonstrate the value of your dataset, have you considered evaluating these models on other existing benchmarks in agriculture (e.g., AgriBench[1] and SeedBench[2]) or even broader scientific benchmarks?
5. In the main text, many references to the appendix are generic (e.g., line 239). Would it be possible to update these to point to the specific section or table in the appendix for easier navigation?
6. The font size in Figure 5 is too small to be legible, even when zoomed in. Could you please provide a more readable version of this figure?

[1] Yutong Zhou and Masahiro Ryo. AgriBench: A Hierarchical Agriculture Benchmark for Multimodal Large Language Models. arXiv preprint arXiv:2412.00465, 2024.

[2] Ying, Jie, et al. "SeedBench: A Multi-task Benchmark for Evaluating Large Language Models in Seed Science." arXiv preprint arXiv:2505.13220 (2025).

---

### Author Response · Authors · 2025-12-03
**Note.**

We thank all reviewers for their feedback. In response to the comments, we are currently enhancing both the dataset and evaluation framework, as well as broadening our analysis to include diverse post-training techniques. We plan to re-submit this work to a future venue.

---

### Note · Authors · 2025-12-03

I have read and agree with the venue's withdrawal policy on behalf of myself and my co-authors.